# Unsupervised meta-clustering identifies risk clusters in acute myeloid leukemia based on clinical and genetic profiles

Jan-Niklas Eckardt [1,2✉], Christoph Röllig[1], Klaus Metzeler [3], Peter Heisig[4], Sebastian Stasik[1], Julia-Annabell Georgi[1], Frank Kroschinsky [1], Friedrich Stölzel[1], Uwe Platzbecker[3], Karsten Spiekermann[5], Utz Krug[6], Jan Braess[7], Dennis Görlich [8], Cristina Sauerland[8], Bernhard Woermann[9], Tobias Herold [5], Wolfgang Hiddemann[5], Carsten Müller-Tidow [10,11], Hubert Serve[12], Claudia D. Baldus [13], Kerstin Schäfer-Eckart[14], Martin Kaufmann[15], Stefan W. Krause [16], Mathias Hänel[17], Wolfgang E. Berdel[18], Christoph Schliemann [18], Jiri Mayer[19], Maher Hanoun [20], Johannes Schetelig [1], Karsten Wendt[2,4], Martin Bornhäuser [1,11,21], Christian Thiede [1] & Jan Moritz Middeke[1,2]

## Abstract

**Background** Increasingly large and complex biomedical data sets challenge conventional hypothesis-driven analytical approaches, however, data-driven unsupervised learning can detect inherent patterns in such data sets.

**Methods** While unsupervised analysis in the medical literature commonly only utilizes a single clustering algorithm for a given data set, we developed a large-scale model with 605 different combinations of target dimensionalities as well as transformation and clustering algorithms and subsequent meta-clustering of individual results. With this model, we investigated a large cohort of 1383 patients from 59 centers in Germany with newly diagnosed acute myeloid leukemia for whom 212 clinical, laboratory, cytogenetic and molecular genetic parameters were available.

**Results** Unsupervised learning identifies four distinct patient clusters, and statistical analysis shows significant differences in rate of complete remissions, event-free, relapse-free and overall survival between the four clusters. In comparison to the standard-of-care hypothesis-driven European Leukemia Net (ELN2017) risk stratification model, we find all three ELN2017 risk categories being represented in all four clusters in varying proportions indicating unappreciated complexity of AML biology in current established risk stratification models. Further, by using assigned clusters as labels we subsequently train a supervised model to validate cluster assignments on a large external multicenter cohort of 664 intensively treated AML patients.

**Conclusions** Dynamic data-driven models are likely more suitable for risk stratification in the context of increasingly complex medical data than rigid hypothesis-driven models to allow for a more personalized treatment allocation and gain novel insights into disease biology.

## Plain language summary

There are various ways in which clinicians can predict the risk of disease progression in patients with leukemia, helping them to treat the patients accordingly. However, these approaches are usually designed by human experts and might not fully capture the complexity of a patient's disease. Here, with a large cohort of patients with acute myeloid leukemia, we design an unsupervised machine learning model – a type of computer model that learns from patterns in data without human input—to separate these patients into subgroups according to risk. We identify four distinct groups which differ with regards to patient genetics, laboratory values, and clinical characteristics. These groups have differences in response to treatment and patient survival, and we validate our findings in another dataset. Our approach might help clinicians to better predict outcomes in patients with leukemia and make decisions on treatment.

A full list of author affiliations appears at the end of the paper.

The ever-growing complexity of medical data poses a challenge for researchers and clinicians alike in determining meaningful parameters that delineate different groups of patients according to disease biology, clinical presentation, and outcome. Acute myeloid leukemia (AML) is a clinically and genetically heterogenous disease and recent studies have established common genetic alterations[1–3] that are used to stratify patients into risk categories which ultimately guide treatment decisions, such as the European Leukemia Net 2017 (ELN2017) recommendations[4]. Commonly, the underlying statistical models looking for a connection between a patient feature and outcome are hypothesis-driven, i. e. they need human-drafted hypotheses for data analysis[5,6]. With the rise of 'big data' in healthcare, finding meaningful patterns in a diverse and high-dimensional data set becomes more and more challenging with this conventional approach[7]. To address this challenge, data mining and machine learning (ML) methods are applied to 'big' medical data sets to find patterns that represent clinically meaningful disease phenotypes[8] with AML being a model disease for the implementation of computational data analysis methods[9]. There is a plethora of different approaches that greatly depend on the data set and the research question, but in general we can distinguish between supervised and unsupervised learning. Briefly, supervised ML requires labeled training data to train the algorithm and subsequently matches fitting labels to unlabeled elements according to patterns derived during the training stage[10]. In contrast, unsupervised ML aims to discover inherent patterns in the data without the need for manually drafted labels and can be used for knowledge retrieval where the algorithm explores similarities and differences between elements in a sample and thereafter groups these elements into clusters based on distinctive underlying patterns[11]. Thus, large-scale complex data sets can provide novel insights into the biology of a variety of malignancies such as AML[12], and unsupervised ML can find structure in these data sets to derive clinically relevant information[13]. Nevertheless, the majority of recent studies applying clustering algorithms in cancer focuses only on genetic data and neglect clinical information. However, the combination of clinical, laboratory and genetic data seems more meaningful to daily practice as it may incentivize clinical decision-making based on commonly available data. From a technological perspective, recent studies have often used only a single clustering method for the analysis of the respective data sets, however, differences in algorithms may lead to different results and it remains unclear whether the usage of different algorithms on the same data sets may yield the same results thereby sparking a discussion of reproducibility of ML-based research in cancer.

In this study, we investigated a cohort of 1383 intensively treated AML patients from 59 hematological centers across Germany according to clinical, laboratory, cytogenetic and molecular genetic data. To ensure the validity of our results, we used different combinations of transformation and unsupervised clustering algorithms. Their individual cluster results were then processed by meta-clustering to find overlapping prognostic patient clusters. Results were then validated on an external cohort of 664 intensively treated AML patients.

## Methods

**Patient data**. 1383 AML patients were retrospectively identified from previously reported multi-center trials (AML96[14], AML2003[15], AML60 + [16], and SORAML[17]) and the German Study Alliance Leukemia (SAL) bioregistry (NCT03188874) encompassing 59 centers specialized in the treatment of hematological neoplasms. Patients ≥18 years with AML diagnosed according to WHO criteria[18] and curative treatment intent were eligible for the purpose of this study. Patients with acute promyelocytic leukemia were excluded. For external validation of an unsupervised learning task, the availability of overlapping features between an original cohort and an external cohort is essential since the distribution of the data shape is tightly connected to the features for any given data set. Hence, a data set that diverges substantially in its available features will inevitably produce different results in an unsupervised learning task. We obtained another large cohort of 664 intensively treated AML patients for whom the same eligibility criteria applied and for whom the same features were available (with the exception only of *IKZF1* and *FLT3*-TKD mutation status). This cohort was provided by the AML Cooperative Group, a multicenter collaborative group of university centers from southern Germany and Austria, and was comprised of patients treated within previously reported clinical trials (AMLCG-1999 and AMLCG2008)[19,20]. Specific treatment regimens for all different trial protocols are summarized in Supplementary Data 1. AML status was defined as de novo when no prior hematologic malignancy was reported. AML was defined as secondary (sAML) when prior myeloid neoplasms such as myelodysplastic syndromes were reported or treatment-related (tAML) when prior exposure to radio- and/or chemotherapy for other malignancies was reported. Complete Remission (CR) was defined according to the ELN2017 recommendations[4]. Sample collection, biobanking, use of samples and clinical information as well as analysis of individual patient data was carried out under the auspices of the SAL bioregistry and the AMLCG. All these activities carried out for the purpose of retrospective research such as this study on previously acquired data were previously approved by the Institutional Review Board of the Technical University Dresden (EK 98032010) and the Institutional Review Board of the Ludwig-Maximillians-University Munich (EK427-13). All participants gave their written informed consent according to the Declaration of Helsinki to having their data used for retrospective research in addition to the individual prospective studies.

Pre-treatment biomaterial from bone marrow aspirates or peripheral blood samples of all patients was screened using high resolution fragment analysis for *FLT3*-ITD[21], *NPM1*[22] and *CEBPA*[23]. Next-Generation Sequencing (NGS) with the Illumina TruSight Myeloid Sequencing Panel was used for additional molecular aberrations. This panel covers 54 genes (Table S1) which are frequently altered in myeloid malignancies as described in detail previously[24,25]. For cytogenetics, standard techniques for chromosome banding and fluorescence-in-situ-hybridization (FISH) were used.

**Data pre-processing and dimensionality reduction**. Multi-center data were merged in a MySQL (Oracle, Austin, TX, USA) database. Features used for cluster generation were available upon initial diagnosis and were either clinical variables (such as age, sex etc.), laboratory variables (such as Hb levels, platelet and white blood cell count etc.) or cytogenetic or molecular genetic alterations comprising a total of 212 parameters. Supplementary Data 2 shows a full list of variables and their frequencies in the patient cohort. Further, it has to be noted that in an unsupervised setting, a number of n features is transformed into $n$ axis of a coordinate systems which represent the model space. To reduce model dimensionality[26], variables that were present in <1% of the patient cohort were excluded from analysis. This is intended to tackle the so-called 'curse-of-dimensionality', where computation is destabilized by adding dimensionality in a data set with a smaller number of samples compared to the available number of features. After excluding sparse features in order to make computations more stable and efficient, 61 features

were left. Importantly, no outcome features such as achievement of CR or survival times (EFS, RFS, and OS) were used for cluster generation. These outcome features were explicitly excluded from cluster generation as they may substantially bias cluster assignments. As individual features represent axis in a coordinate system, weights between features are uniformly equal and cannot be modified externally, i. e. each feature for cluster generation is as important as any other feature. Modifying features, either by inclusion of novel features or exclusion of present features, will therefore obviously modify the shape of the intermediary model and hence the results. Nominal and ordinal variables were one-hot encoded. Continuous variables were standardized to the z-score. Many statistical and machine learning models do not cope well with large amounts of missing data which may ultimately result in unstable or biased models depending on the mechanisms of missingness. A full list of absolute and relative numbers of missing values for the features included in the model is provided in Supplementary Data 3. Since unsupervised clustering itself poses a 'black blox'-like dilemma with regard to explainability, introducing a multiple imputation mechanism that generates different results each time an imputation is run would not be a suitable option, if one aims at reproducible results. No missing data was present for age, sex and molecular alterations (with the exception of subtyping for CEBPA mutations into specific domains in 31 cases), and only a fraction of karyotypes (5.9%) and laboratory values (range 0.2 – 7.4%) were missing. In order to generate reproducible outputs for imputation that would still be solid in k-fold cross-validation and potential re-runs of the model, continuous variables were imputed with the median of the respective variable. Missing categorical variables were tagged as unknown. Unknown variables were dropped at the pre-processing stage. For example, a patient with all data available except for information on extramedullary AML manifestations will still be included in clustering, however will be represented by a feature vector consisting of 60 values rather than 61 in a 61-dimensional intermediary model space. With regard to outcome variables, which were strictly withheld from cluster generation, only cases with available survival times were used in Kaplan-Meier analysis (see numbers-at-risk-tables, Fig. 4). There was no imputation of missing outcome variables.

Since scaling and transformation of such a heterogenous data set could potentially lead to distortions of individual relations between elements, the individual elements were transferred to a model space of reduced dimensionality while at the same time retaining their original similarities expressed in Euclidean distances and therefore allowing for accurate clustering of elements and ensuring model stability by reduced dimensionality. The transfer between model space and original space was interchangeably available so that re-transformation to original scales (i. e. mmol/l instead of z-standardized values) for clinically meaningful interpretation of variables was possible. As no uniformly optimal algorithm for transformation and dimension reduction exists, but rather an optimal strategy has to be determined based on the given data set, we used eleven transformation algorithms listed in Table S2. A crucial adjustment is the number of targeted uniform dimensions in the model space, as this substantially influences the shape and expressiveness of the intermediate model. A high number of uniform dimensions allows for accurate mapping but makes it difficult for the subsequent clustering methods to detect distinct clusters, while a low number possibly distorts the original representation but allows for more distinctive clustering.

**Unsupervised learning and meta-clustering**. On the basis of the transformed data models, unsupervised clustering was performed.

Clustering was designed according to the following rules to guarantee: (i) a minimum size of clusters (not smaller than 10% of the entire cohort), i. e. feature abstraction should not be too strong or too weak which would result in a large number of clusters consisting of only a few patients each; (ii) large differences between clusters, i. e. the patients in different clusters should differ maximally in terms of their features and; (iii) small differences within clusters, i. e. the patients within a cluster should differ minimally in terms of their features. Ideally, this would allow for clusters that are in themselves maximally homogenous but in comparison to another cluster maximally heterogenous. Since, again, no optimal algorithm exists and performance has to be evaluated based on the given data, we tested a variety of combinations: (i) For data transformation and mapping, eleven different algorithms were used (Table S2); (ii) data was transformed to a target dimensionality pre-determined at each run ranging from 2 to 6 ($n = 5$); (iii) then, eleven different clustering algorithms (Table S3) were employed to assign patients to groups based on outcome. In total, we considered 605 different possible clustering combinations in a large-scale grid search. To define a starting value for grid search in the model space and make the behavior of pseudo-stochastic methods repeatable, a random seed was used, and to reduce the influence of pseudo-randomness, each run was repeated for 10 iterations.

To ensure optimal quality of the clusters, a sanity-check was performed after each clustering. This is intended to make clusters both interpretable and clinically meaningful. From a purely mathematical standpoint, it would be conceivable that cluster generation would end in 1383 clusters, i. e. one cluster for each patient, or one cluster for all patients as well as any number of clusters in between. As this is not practicable to be used in the clinical routine, a set of rules has to be introduced to ensure that generated clusters fulfill a set of criteria that enables their transferability from computer to bedside. To highlight biological differences between AML patients, clusters have to differ maximally in their data distribution. In an n-dimensional space, this means that each patient is represented by a point in a coordinate system with n axis. Cluster generation should then be able to delineate clusters by maximizing the distance between patients in the n-dimensional space. This means that only patients that are close to each other with regard to their features should be considered belonging to the same cluster while patients that are different, i. e. are farther apart in this n-dimensional space, should belong to a different cluster. Further, clusters should be of adequate size to be meaningful in clinical routine. A cluster that consists of only e. g. five patients (of 1383 patients in total) would not be meaningful. At the same time, such small but distinctive clusters would dramatically increase the overall number of clusters and thereby further hinder clinical applicability as it seems rather improbable that a clinician would utilize (or even memorize) a risk assessment tool consisting of a two- or even three-digit number of individual risk groups. Therefore, cluster sizes and numbers of clusters (minimum of 10% of overall patients, i. e. limiting cluster numbers to a maximum of 10) were also considered as rules in the sanity check. Since we aimed to include a variety of transformation and clustering algorithms rather than subjectively selecting any specific combinations, it has to be pointed out that depending on the distribution of the data not all transformation algorithms are suitable to work with all kinds of data and not all combinations of transformation and clustering algorithms work well with each other. As our explicit goal from a technological perspective was to manually interfere as little as possible and rather let the model decide for itself and eliminate unfitting outliers, we nevertheless allowed all 11 ×11 combinations of transformation and clustering algorithms. Conceivable, a dysfunctional match-up between any of these

algorithms would produce a result that would be extreme in some sense, e. g. would produce to large or small, to few or to many clusters, or patients within the clusters would be too different and patients between the clusters would be too similar. Again, this is where the sanity check comes in that eliminates such extreme and mathematically unsound combinations before a final analysis is undertaken. Thereby, we do not need to sort through combinatorial outputs manually and potentially introduce subjectivity, but rather let the model decide under the pretense of the above-specified rules for clustering which clusters and thereby which algorithmic combinations satisfy the pre-specified quality criteria. It needs to be stressed, that the sanity check did in no way interfere with feature selection nor did it include any outcome variables to maximize cross-cluster heterogeneity. Outcome variables were strictly withheld from cluster generation and only statistically analyzed after final clusters were generated. If an individual run fails the sanity-check, for example if it includes clusters harboring only 1 patient each, the result is discarded and not evaluated further.

Inherently to unsupervised learning, different methods of clustering, i. e. different algorithms, will result in different outputs. Hence, no single 'best' algorithm exists for any given task, but rather the algorithm has to be evaluated within the context of a specific data set. What further complicates this issue, is that there are no labels in unsupervised learning. A supervised learning task is trivial to evaluate with regard to differential model performance: A given set of robust ground truth labels are provided and model performance is simply evaluated on how well each single model predicts the previously unseen labels. Then, the best model is the one with the highest hit-rate between predicted and ground truth labels. Since there are no labels (and often no real ground truth) in unsupervised learning, a selection of the 'best' algorithm is rather subjective. In our set-up of 605 different possible combinations, one could conduct statistical analysis for each combination, but would still be left with the issue of what output would be considered the 'winner'. As this fundamentally ends in a subjective and therefore potentially biased manual decision, we used meta-clustering not to select the 'best' clustering output, but rather to average all valid (after the sanity check) outputs. Still, to make individual algorithmic combinations numerically comparable, we used the silhouette coefficient[27], the Calinski-Harabasz-Index[28], and the Davies-Bouldin-Score[29]. Silhouette analysis[27] measures how close a point in a cluster is to points in neighboring clusters. On a scale from $-1$ to $+1$, samples that are far away from neighboring clusters will receive a value close to $+1$ while a value close to 0 means that a sample is close to the decision boundary between two clusters whereas a value close to $-1$ indicates an error in cluster assignment. The Calinski-Harabasz-Index (also known as the Variance Ratio Criterion)[28] is the ratio between the sum of inter-cluster dispersion and the sum of intra-cluster dispersion. It ranges from 0 to (theoretically) infinity with higher values indicating higher clustering quality. Lastly, the Davies-Bouldin-Score[29] is an average similarity index that compares each cluster to its most similar cluster. A ratio is formed of intra-cluster distances to inter-cluster distances. The score ranges from 0 to (theoretically) infinity where values close to 0 indicate a higher distance between clusters and thus, better overall cluster quality.

Meta-clustering was performed using principal component analysis for transformation and mean shift for clustering. Meta-clustering essentially does not cluster the raw data, but clusters the output of the previous algorithms (Fig. 1). Thereby, the individual outputs of all 605 combinations are used to generate clusters based on similar cluster assignments, i.e. patients that are farther apart in the n-dimensional model space for the majority of clustering algorithms will also be farther apart in meta-clustering

(on average) and will therefore be put in two different final clusters. In that way, the chance for false positives or negatives that may be given when using only one combination of transformation and clustering algorithms is reduced and the need for human judgment of what makes one combination better or worse than another (and thereby potentially introduce bias) is limited. Code was generated using Python 3.8 (Python Software Foundation, Fredericksburg, Virginia, USA). Python packages that were used are summarized in Table S4.

**External validation as a supervised learning task**. Unsupervised learning inherently only sorts samples into clusters. Therefore, any addition of new data will lead to a completely new sorting. Ideally, original clusters would be retained, however, this depends on the distribution of the newly introduced data. In order to externally validate our model, a mere addition of data is therefore insufficient. Unsupervised learning does not consider labels as there is no evident ground truth, but rather commonalities and differences between samples are at the center of the analysis. Contrastingly in supervised learning, labels can be used to learn features that distinguish a certain set of specimens (e. g. shape and color can be learned to distinguish apples and bananas). Hence in our set-up cluster, assignment can be viewed as labels that are determined by patient features. Therefore, in order to include new data into the model, the set-up can be modified to a supervised learning task as soon as one set of final cluster assignment labels has been generated previously. Potentially, this process can be iterated ad infinitum: Use unsupervised clustering for cohort A, predict cluster assignments based on A's features for cohort B. It could also be modified in order to alter the original cluster results: Use unsupervised clustering for cohort $A + B$, predict cluster assignments based on A and B's features for cohort C etc. For this design to function properly, the overlap in available features between cohort A and B has to be high (ideally fully matching). We obtained an external cohort of 664 intensively treated AML patients from previous clinical trials as described above. Cluster assignment labels were learned using supervised learning on the original cohort and subsequently, cluster assignment was predicted on the external cohort in order to sort them within the existing clusters. Pre-processing for the external data did not differ from our original cohort as described above. Again, there is no universally 'best' algorithm for such a supervised learning problem. Therefore, different algorithms have to be evaluated based on their individual performance for a given task. To do this, we used a train-test-split on the original cohort of 80:20 and evaluated four different supervised algorithms: naïve Bayes, gradient boosting, random forest and logistic regression. Algorithm performance was evaluated using AUROC, precision, recall, and F1-score. The overall best performing supervised algorithm was selected in order to assign cluster labels to the external validation cohort. Since there is no ground truth for cluster labels with regard to the external cohort, only test set performance on the orginal cohort can be reported. Based on these cluster assignments, survival analysis was performed on the external cohort and compared to our original cohort.

**Statistical analysis**. Statistical significance was determined using a significance level α of 0.05. All tests were carried out as two-sided tests. Univariate analysis for binary outcomes including complete remission (CR) was carried out using logistic regression models to obtain odds ratios (ORs) in comparison to the overall sample. For survival analysis including the evaluation of event-free survival (EFS), relapse-free survival (RFS) and overall survival (OS), the Kaplan-Meier method and the log-rank test were used. For univariable and multivariable analysis regarding

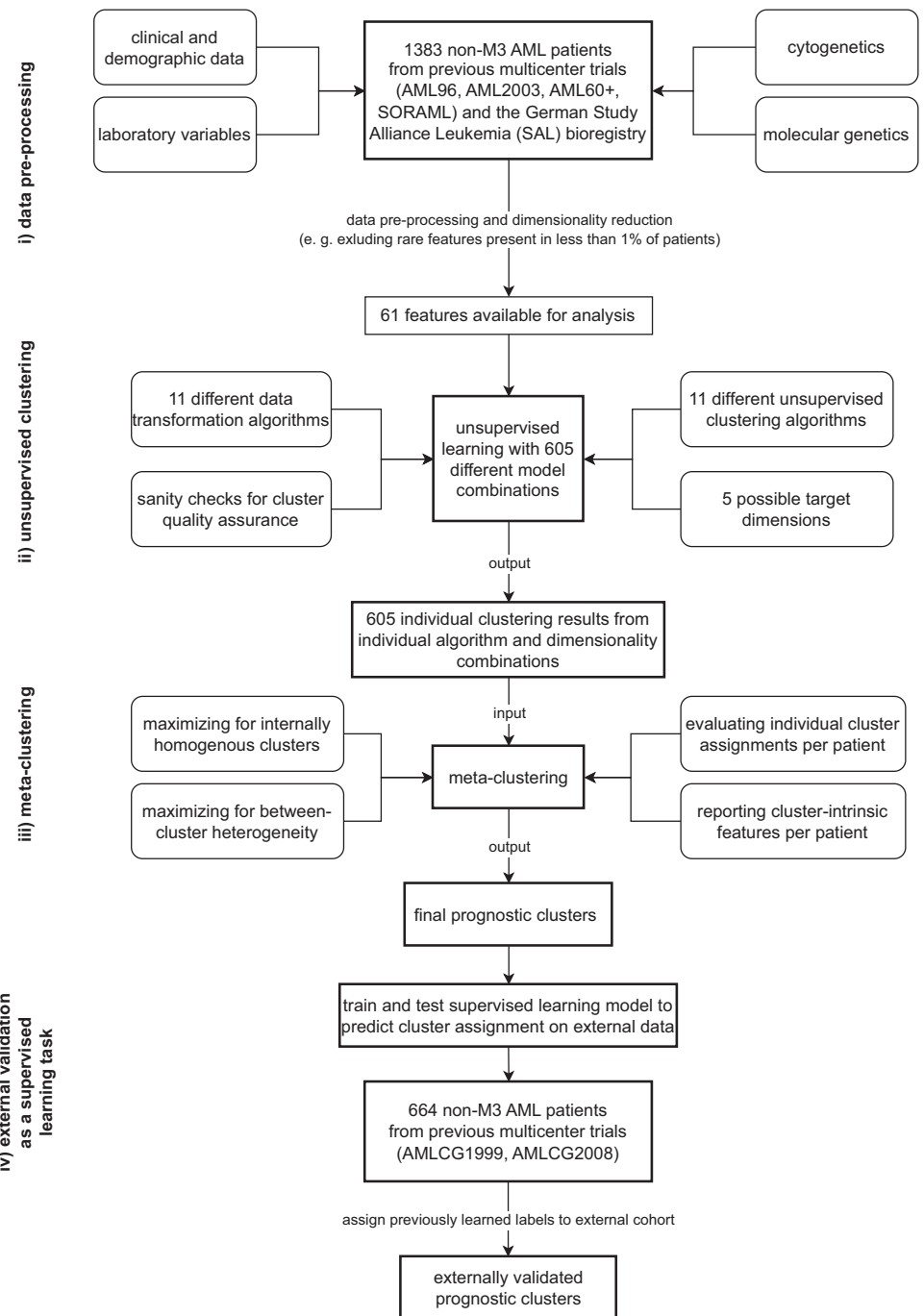

**Fig. 1 Step-wise workflow of unsupervised learning.** After pre-processing of multimodal patient data (i), unsupervised learning (ii) was performed with different combinations between target dimensionality (2–6, $n = 5$), data transformation ($n = 11$) and unsupervised clustering algorithms ($n = 11$). The individual outputs of each algorithm combination were gathered and uses as input for meta-clustering (iii) to find patient clusters that in themselves are maximally homogenous while at the same time differ maximally from other patient clusters. Thus, final clusters are identified and individual features of patients within these clusters become available for further analysis. In the next step, the previous cluster assignments can be treated as labels for supervised learning (iv). After training and testing on the original cohort, the highest performing classifier is being selected for assigning cluster labels to an external validation cohort.

survival, Cox-proportional hazard models were used to obtain hazard ratios (HRs). For both ORs and HRs, 95%-confidence intervals (95%-CI) are reported. Differences between clusters with respect to categorical variables were evaluated using Fisher's exact test, and continuous variables were compared using ANOVA, if the assumption for normality was met. Normality was evaluated using the Shapiro-Wilk test. If the assumption of normality was violated, the Kruskal-Wallis test was used. The Benjamini-Hochberg method[30] was used to adjust for multiple testing with regard to the respective outcome variables. Statistical analysis was performed using STATA BE 17.0 (Stata Corp, College Station, TX, USA).

**Reporting summary**. Further information on research design is available in the Nature Portfolio Reporting Summary linked to this article.

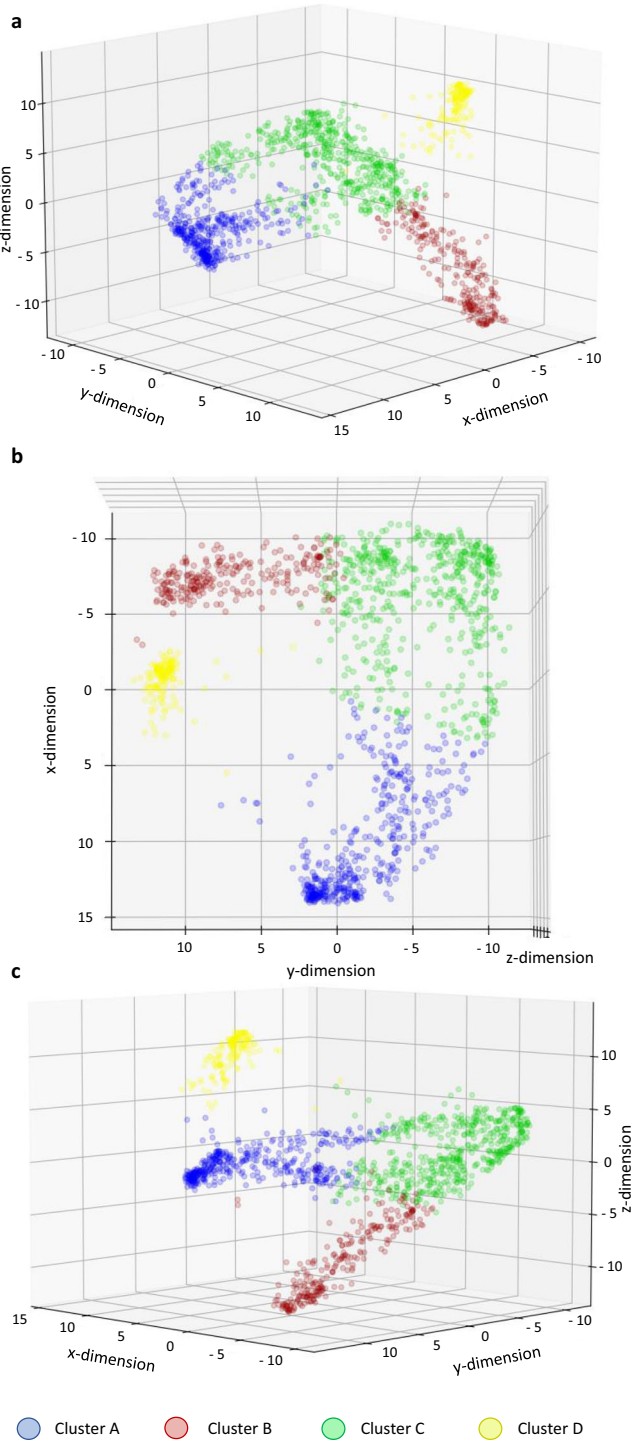

**Fig. 2 Three-dimensional feature space with final meta-clustering output.** Meta-clustering was used to aggregate individual clustering results and the output was mapped to a three-dimensional (x, y, z) feature space depicted from three different angles (**a** front view, **b** top view, **c** side view). Four final meta-clusters were obtained. Each dot represents a single patient ($n = 1383$). Colors indicate patient cluster assignments. Axes represent deviations from the mean feature vector value on a standardized linear scale. Since features can be represented as vectors, for each patient a mean feature vector (i. e. a vector that aggregates all 61 features) can be calculated that determines the patient's location in the feature space where a value of 0 indicates the mean for the entire cohort and values larger or smaller than 0 correspond to deviations from the average. Thus, patients can be separated in the feature space based on different expressions of their feature vectors which allows for subsequent (meta-)clustering.

## Results

**Unsupervised learning identifies four distinct clusters of AML patients.** Using meta-clustering, we aggregated the results of 605 different combinations of target dimensionalities, data transformation and clustering algorithms and obtained four final clusters that were mapped to a three-dimensional space (Fig. 2). Without any manual feature selection, 61 patient variables (Supplementary Data 4) were used for cluster formation as sparse and redundant variables were excluded by machine learning for dimensionality reduction. These features were solely comprised of clinical variables such as age, AML status (de novo, sAML, tAML) etc., laboratory variables such as bone marrow blast count or white blood cell count etc. upon initial diagnosis, and molecular and cytogenetic variables. Individual clustering performance of different combinations of transformation and clustering algorithms as well as target dimensions were evaluated using the silhouette coefficient[27], the Calinski-Harabasz-Index[28], and the Davies-Bouldin-Score[29]. Silhouette coefficients, Calinski-Harabasz-Indices and Davies-Bouldin-Scores ranged from −0.12 to 0.72 (closer to 1 is better), 4.56 to 3533.94 (higher is better), and 0.46 to 18.46 (closer to 0 is better), respectively. 517 of 605 possible algorithmic combinations passed the sanity check. Their individual performance metrics can be viewed in Supplementary Data 5. Since a good numerical value in the above-mentioned indices does not necessarily guarantee meaningful knowledge retrieval from clustering and picking a 'winner' combination still remains somewhat subjective, meta-clustering was used to average the results of combinations that passed the sanity check.

Supplementary Data 6 shows a detailed comparison of baseline patient characteristics and significant differences between clusters. With respect to baseline patient characteristics, patients assigned to cluster A, C, and D predominantly had de novo AML, while cluster B had equal proportions of patients with de novo and sAML. We found significant differences in age and sex between the four clusters with patients in cluster D being the youngest (median age 49 years) and patients in cluster B being the oldest (median age 60.5 years), while cluster A harbored the largest proportion of female patients (59.7%) and cluster C the largest proportion of male patients (66.0%). Regarding the FAB-classification[31], cluster A showed the largest proportion of AML-M5 and B harbored the largest proportion of AML-M6 while at the same time having the smallest proportion of AML-M1. Cluster C had the highest proportion of AML-M4, while D consisted predominantly of AML-M1 and -M2. With regard to laboratory values at initial diagnosis, patients in cluster A showed the highest white blood cell count (median: $52.5 * 10^9$/l, Fig. S1A), highest platelet count (median: $62 * 10^9$/l, Fig. S1B), highest LDH (median: 613 U/l, Fig. S1C), and highest bone marrow blast count (median: 77%, Fig. S1D). Patients in cluster B had the lowest white blood cell count (median: $4.0 * 10^9$/l, Fig. S1A), lowest LDH (median: 293 U/l, Fig. S1C), and lowest peripheral blood blast counts (median: 10%, Fig. S1E). Lastly, patients in cluster C showed the lowest bone marrow blast counts (median: 34%, Fig. S1D), while patients in cluster D had the lowest platelet counts (median: $39 * 10^9$/l, Fig. S1B). There was no significant difference in hemoglobin levels between the clusters (Fig. S1F).

**Clusters identified by unsupervised learning differ according to molecular and cytogenetic alterations.** The four clusters showed distinct differences in the expression of molecular and cytogenetic alterations displayed in Fig. 3. Cluster A had a high proportion of

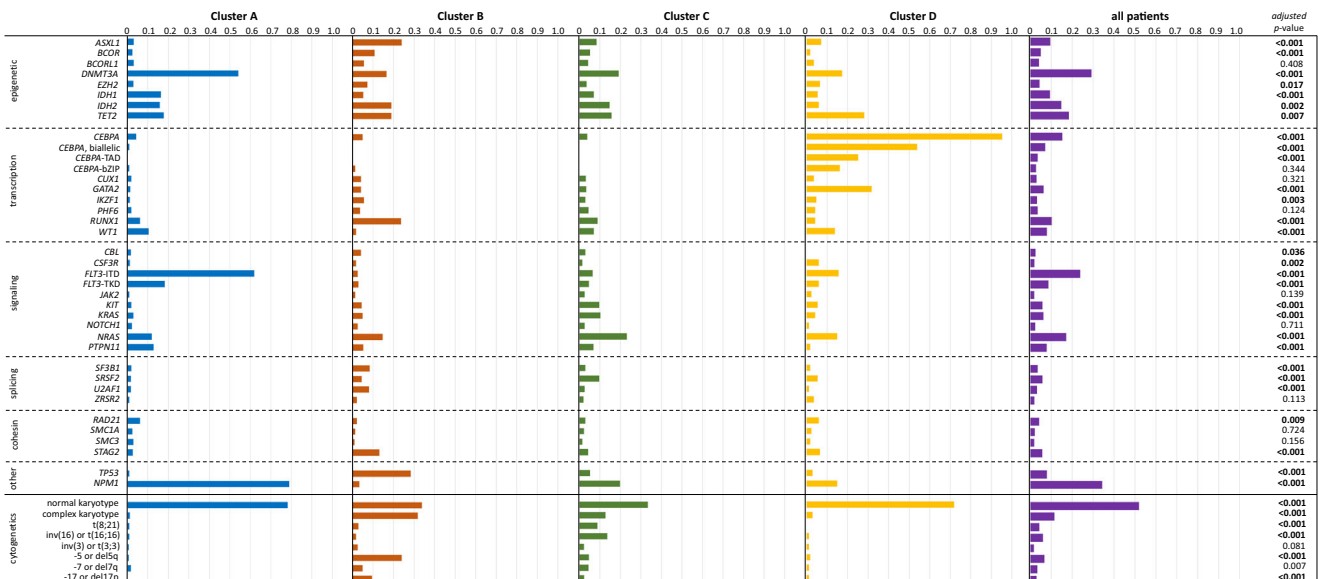

**Fig. 3 Differences in molecular genetics and cytogenetics between clusters.** Final clusters identified by meta-clustering differed significantly regarding their molecular and cytogenetic composition. $n_{Cluster\ A}$ = 424 patients, $n_{Cluster\ B}$ = 256 patients, $n_{Cluster\ C}$ = 536 patients, $n_{Cluster\ D}$ = 167 patients.

normal karyotypes (77.6%) while complex karyotypes were rare (0.7%). The majority of patients in A showed mutations of *NPM1* (78.3%), *FLT3*-ITD (61.3%), and *DNMT3A* (53.5%). Further, *FLT3*-TKD (17.7%), mutations of *IDH2* (17.2%), *TET2* (17.2%), *IDH1* (15.8%), *PTPN11* (12.3%), and *NRAS* (11.3%) were present. Cluster B had equal proportions of normal (33.2%) and complex karyotypes (31.3%) and a high proportion of patients with −5 or del(5q) (23.4%). The most frequent molecular aberrations in B were *TP53* (27.7%), *ASXL1* (23.4%), *RUNX1* (23.0%), *TET2* (18.4%), *IDH2* (18.4%), *DNMT3A* (16.0%), *NRAS* (14.1%), *STAG2* (12.5%), and *BCOR* (10.2%). With respect to cytogenetic aberrations, cluster C showed a high proportion of normal karyotypes (32.8%) while also harboring patients with complex karyotypes (12.1%). Furthermore, 13.1% of patients in C had inv(16) or t(16;16) and 8.2% had t(8;21). Alterations of *NRAS* (22.6%) were the most frequent molecular alteration in cluster C followed by mutated *NPM1* (19.2%), *DNTM3A* (18.7%), *TET2* (15.1%), and *IDH2* (14.2%). Lastly, cluster D consisted of a majority of patients with normal karyotypes (71.3%). Regarding molecular alterations, D was dominated by mutated *CEBPA* (94.6%). The majority of patients carried biallelic *CEPBA* mutations (53.3%) with 24.6% of patients carrying mutations in the TAD-domain and 15.6% of patients carrying mutations in the bZIP-domain only. Other frequent alterations were *GATA2* (31.1%), *TET2* (27,5%), *DNTM3A* (16.8%), *FLT3*-ITD (15.0%), *NRAS* (14.4%), *NPM1* (14.4%) and *WT1* (13.2%). Specifically with respect to *FLT3*-ITD ratio, we found significant differences between clusters (adj. *p* < 0.001) with cluster A having the largest median ratio of 0.675 while the remaining clusters B, C, and D had median ratios below 0.5 (0.425, 0.255, and 0.220 respectively, Fig. S2). Supplementary Data 7 shows all absolute numbers and proportions of molecular and cytogenetic alterations in the different clusters in detail. The connections between individual aberrations per cluster are displayed as a heatmap in Fig. S3.

**Clusters differ in response to induction therapy and long-term survival.** All patients in our cohort received intensive induction therapy. 61.4% of patients received double induction therapy (*n* = 849) while 38.6% of patients (*n* = 534) received only a single course of induction therapy. Allogeneic hematopoietic stem cell transplantation (HSCT) status was explicitly not used as a feature

for cluster generation as it may have substantially biased results with regard to survival times. The rate of HSCT either upfront or as salvage treatment did not differ significantly across clusters (adj. *p* = 0.058 and adj. *p* = 0.675, respectively). Therefore, any potential effect of HSCT (in total rather than regarding HSCT in first CR) on cluster outcome appeared to be equally distributed among clusters rather than distorting outcomes for any cluster in particular. With respect to achievement of CR, patients in cluster D followed by A had a significantly increased OR of 2.42 and 1.60 compared to the overall sample, respectively, while B showed a significantly decreased OR of 0.36. Median EFS was significantly increased for D with 11.2 months (HR = 0.74) followed by C with 9.1 months (HR = 0.83), while B showed the lowest EFS with 1.9 months and a corresponding HR of 1.84 (Fig. 4A). Patients in cluster A and B showed significantly increased HRs for relapse of 1.25 and 1.39, respectively, with a correspondingly decreased RFS of 14.8 and 11.9 months, respectively. In contrast, cluster C showed a significantly decreased HR of 0.73 for relapse with a corresponding median RFS of 30.7 months while no significant differences were found for cluster D (Fig. 4B). OS was significantly increased for cluster D and C with a median of 54.0 and 25.8 months corresponding to HRs of 0.68 and 0.82, respectively, while B had significantly decreased OS with a median of 10.5 months (Fig. 4C) and a corresponding HR of 1.69. Supplementary Data 8 provides a detailed overview of survival times, ORs, HRs and significance levels for each cluster.

**Unsupervised clustering re-stratifies patients compared to the ELN2017 classification.** Across the clusters, the distribution of ELN2017 risk groups differed significantly (ELN2017 favorable, adj. *p* < 0.001; ELN2017 intermediate, adj. *p* = 0.007; ELN2017 adverse, adj. *p* < 0.001). In cluster A, patients were most frequently assigned to the ELN2017 favorable risk group followed by ELN2017 intermediate risk while ELN2017 adverse risk allocation was rare (Fig. 5A). In contrast, B was primarily composed of patients within the ELN2017 adverse or intermediate stratum while a minority were assigned to ELN2017 favorable risk (Fig. 5B). C consisted of equal parts of patients within the ELN2017 favorable and intermediate stratum with a fifth of patients being assigned to the ELN2017 adverse risk group (Fig. 5C). Lastly, D was predominated by patients assigned to the

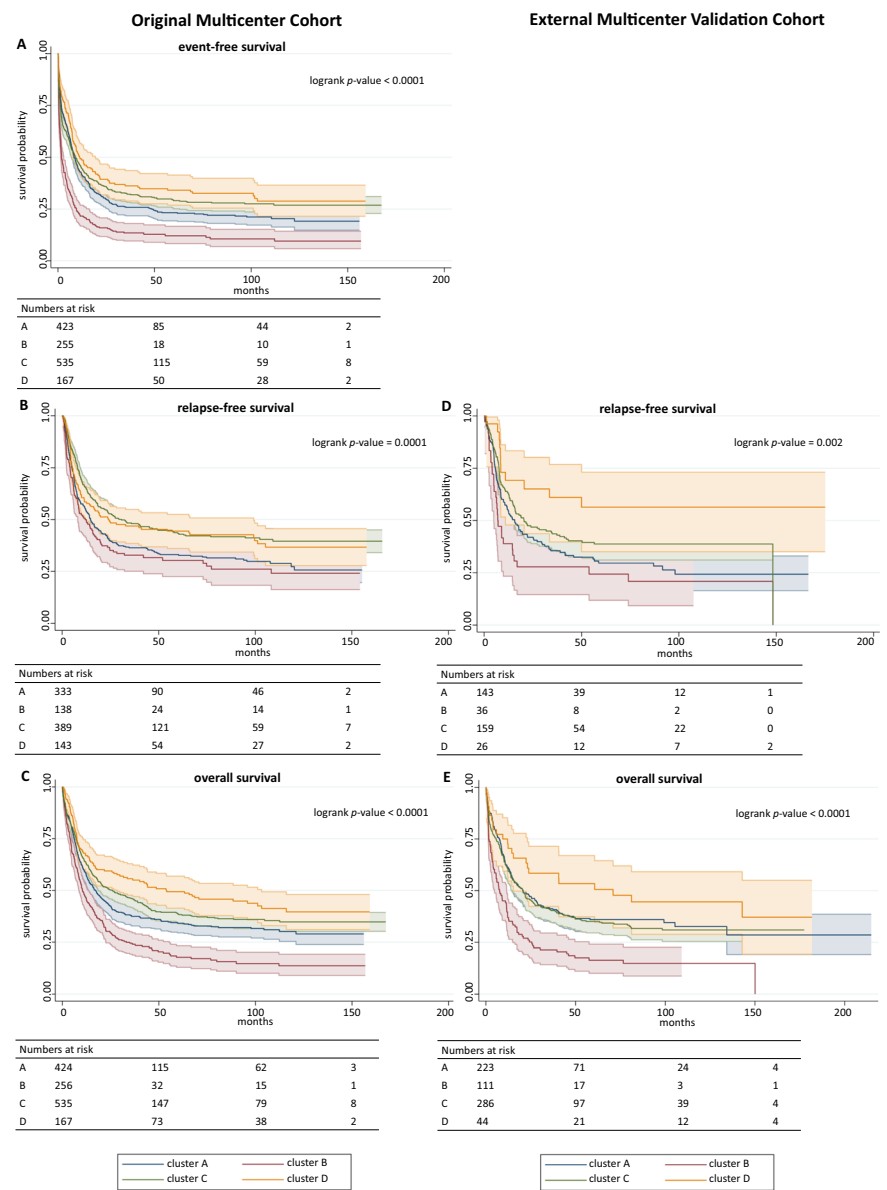

**Fig. 4 Differences in survival times between clusters.** Survival times with regard to event-free (EFS), relapse-free (RFS) and overall survival (OS) were compared using the Kaplan-Meier-method and log-rank test. Results for the original cohort are shown in panels **A**–**C**. These differed significantly between the four clusters: While cluster D showed relatively favorable outcomes followed by cluster C, contrastingly, B was characterized by poor long-term survival followed by cluster A. This is most evident regarding EFS (panel **A**) and OS (panel **C**) while in RFS (panel **B**), both clusters A and B as well as C and D were largely overlapping. For the external validation cohort, no information regarding EFS were available. With respect to RFS (panel **D**) and OS (panel **E**), clusters again showed significantly different outcomes with cluster D bearing the most favorable and cluster B bearing the least favorable outcome. Numbers at risk are shown for the respective time points (0, 50, 100, 150 months). Colored bands represent 95%-confidence intervals. $n_{Cluster\ A} = 424$ patients, $n_{Cluster\ B} = 256$ patients, $n_{Cluster\ C} = 536$ patients, $n_{Cluster\ D} = 167$ patients.

ELN2017 favorable risk group with roughly a quarter of patients within the ELN2017 intermediate stratum and only a small proportion of patients with ELN2017 adverse risk (Fig. 5D). For patients allocated to the ELN2017 favorable risk group within cluster A, 54.9% of patients were bearing mutated *DNMT3A*, while mutated *PTPN11* was found in 20.7% of patients. Mutated *IDH1*, *IDH2* and *NRAS* was found in 19.7%. For patients with ELN2017 intermediate risk in A, mutated *DNMT3A* was seen in 55.7% of patients (Supplementary Data 9). With respect to cluster B, patients in the ELN2017 intermediate subgroup frequently showed mutations of *IDH2* (29.3%), *STAG2* (20.7%), *TET2* (20.7%) as well as *DNMT3A* (23.3%, Supplementary Data 10). Within cluster C (Supplementary Data 11), we observed 95.8% of patients with ELN2017 intermediate risk to bear mutated *CEBPA*

−37.5% of patients with mutations in the TAD-domain, 47.9% in the bZIP-domain, and 8.3% with biallelic mutations – while 35.4% also had mutated *TET2* and 20.8% had mutated *DNMT3A*. Lastly for the five patients with ELN2017 adverse risk within cluster D (Supplementary Data 12), all of the five patients had mutated *CEBPA*, four of which had mutations in the TAD-domain and one had a mutation in the bZIP-domain. All of these patients had concomitant mutations in *RUNX1*.

**Cluster outcomes are preserved in an external multicenter validation cohort**. In stark contrast to supervised learning, external validation of an unsupervised learning task is much harder to accomplish since there is no obvious ground truth an

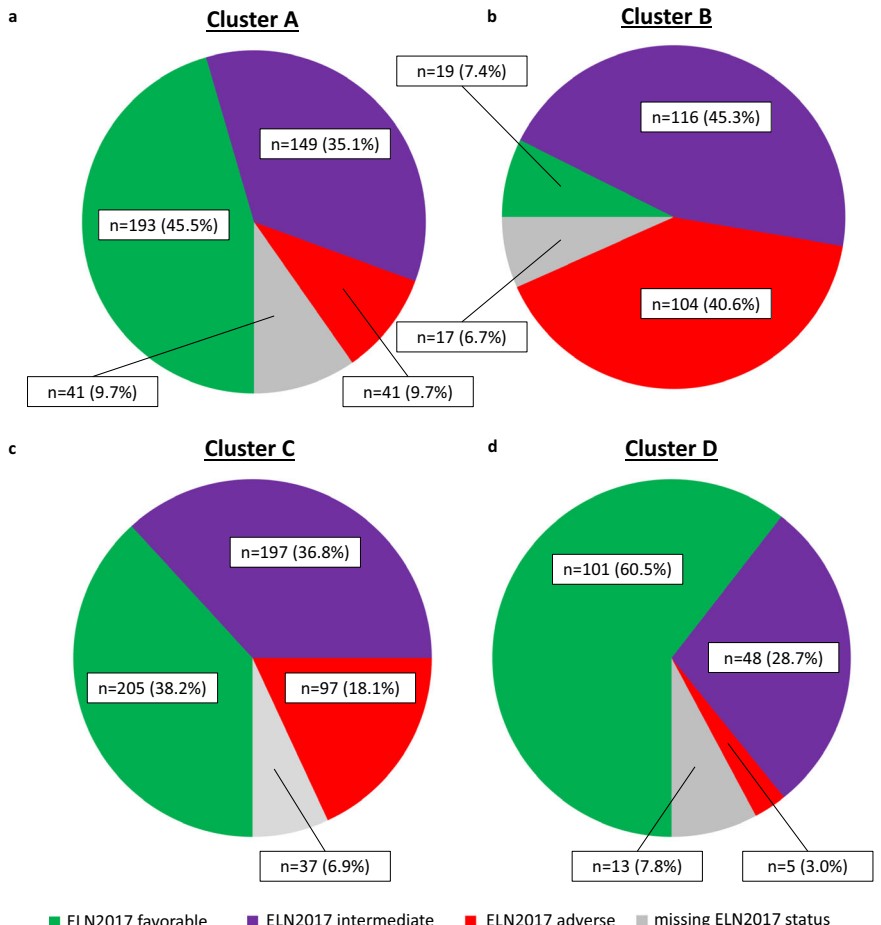

**Fig. 5 Distribution of ELN2017 risk categories across the clusters in the original multicenter cohort.** In each of the four clusters, all three ELN2017 risk groups were represented in differing proportions. Both clusters A (panel **a**) and C (panel **c**) harbored equal proportions of patients assigned to the ELN2017 favorable and intermediate categories, while B (panel **b**) had equal proportions of patients with ELN2017 intermediate and adverse features and D (panel **d**) was dominated by ELN2017 favorable markers, however, followed by the ELN2017 intermediate group. $n_{Cluster\ A} = 424$ patients, $n_{Cluster\ B} = 256$ patients, $n_{Cluster\ C} = 536$ patients, $n_{Cluster\ D} = 167$ patients.

algorithm's performance can be compared against. Furthermore, there is no 'off-the-shelf'-solution to incorporate new data into a previously established cluster without altering the cluster itself. In order to retain previously generated clusters and at the same time add new external data, we transformed the task into a supervised one by using the cluster assignments as learnable labels. We then trained four different supervised algorithms to predict cluster assignments on the original cohort with a 80:20 train-test-split in order to identify the most suitable algorithm for the task. Individual model performance is displayed in Table S5. With a test set AUROC of 0.99, logistic regression was the highest performing algorithm in cluster assignment. Hence, logistic regression was used to assign cluster labels to an external cohort. This cohort was comprised of 664 intensively treated AML patients from previous trials of the Acute Myeloid Leukemia Cooperative Group (AMLCG). In comparison to the original cohort, cluster sizes for each of the four clusters were similar. In general, the directionality of effects, i. e. cluster B showed overall a high risk-profile while cluster D had more favorable outcomes with clusters A and C in between, was retained. As in the original cohort, patients assigned to cluster A were significantly more likely to achieve CR (OR = 2.08). Patients assigned to B again had dismal outcomes as they were significantly less likely to achieve CR (OR = 0.33) and had a significantly decreased RFS and OS as in the original cohort with a HR of 1.68 and 1.87, respectively. C,

again, showed intermediate outcomes with regard to survival times. Lastly, patients assigned to D had the most favorable outcomes with significantly increased RFS and OS (HR = 0.47 and 0.64, respectively) as in the original cohort, while CR rate did not differ in contrast to the original cohort. Detailed information on outcomes in comparison to the original cohort can be taken from Supplementary Data 7 and Fig. 4D, E. Information on EFS was not available for the external validation cohort.

## Discussion
Based on 605 different combinations of target dimensionalities, transformation and clustering algorithms, unsupervised meta-clustering identified four distinct clusters in a large multicenter cohort of 1383 intensively treated AML patients differing according to clinical phenotypes, laboratory parameters, molecular as well as cytogenetic alterations. Results were externally validated in a cohort of 664 intensively treated AML patients. The lessons from this study are threefold in terms of (i) acknowledging disease biology and interconnectedness of genetic alterations in a combinatorial model, (ii) providing a data-driven tool for cohort-based risk assessment with clinically meaningful differences in patient outcome that potentially warrant different disease managing strategies, and thereby (iii) challenging conventional approaches for risk stratification that are expert-opinion-based rather than data driven.

AML biology is more complex than can be acknowledged by mere presence or absence of one marker of favorable or adverse prognosis. Conventional risk assessment tools like ELN2017[4] essentially act as evidence-based and expert-opinion-guided decision trees. The general rule is that as soon as one item on a checklist is present, a certain group assignment (favorable, intermediate, adverse) is undertaken as long as there are no contradicting items for a given patient. For example: A patient has a feature; This feature is considered high-risk in the ELN model; There are no contradicting other features; The patient is categorized as high risk. In this regard, with the exception of a few rules on specific mutation types and co-occurring mutations, genetic alterations within a certain risk group are weighed equally although this contradicts biological rationale with respect to molecular mechanisms. Further, in such a decision tree model, the presence of one alteration in a given risk group, e. g. mutated *RUNX1* in the ELN2017 adverse risk category, fundamentally dictates patient assignment to this risk group without acknowledging further adverse risk factors that may be present for this patient. For example, a patient with mutated *RUNX1* only is treated the same as a patient with a complex karyotype, mutated *TP53* and mutated *RUNX1*. This lack of granularity fails to acknowledge biological differences between genetic alterations within the same risk category and sometimes even across categories within the ELN2017 risk stratification. Results of unsupervised clustering approaches could be one way to perceive that AML biology is much more complex than what can be captured with a simple 'absent vs. present feature'-type of decision tree. In all of our four clusters, patients from all ELN2017 risk categories are represented suggesting a differential outcome for patients with a given mutation in context with co-occurring mutations. Our approach therefore treats available patient information in an interconnected manner in the context of other patients with similar profiles rather than a decision-tree where even one piece of information may be sufficient to fulfill the checklist for a certain group assignment thereby acknowledging implicit rules in the data that are not captured in previous models such as the ELN recommendations.

Cluster assignment was performed using clinical, laboratory and genetic information on the patient level. Individual patient outcome, i.e. achievement of CR or survival times, were explicitly excluded from cluster formation. Thereby, cluster assignment could be used to analyze clusters with regard to treatment response and survival based on their upfront clinical and genetic information. With respect to clinically relevant endpoints such as CR and OS, clusters ranged from favorable prognosis (D) over intermediate (C) to high-risk patients (B) and a hybrid group of patients with increased odds to achieve CR after induction therapy, but contrastingly decreased survival (A). Regarding the latter, cluster A was dominated by normal karyotypes, mutated *NPM1*, *DNMT3A* and *FLT3*-ITD with a ratio >0.5 and accordingly patients were mainly allocated to either the ELN2017 favorable or intermediate risk groups[4]. While the interplay of mutated *NPM1* as well as *FLT3*-ITD depending on its allelic ratio is well characterized[1,21,22,32], the prognostic impact of *DNMT3A* is still controversial as previous studies showed conflicting results reporting both decreased, increased or no impact on survival[3,33,34]. Considering that the rates of patients receiving HCT between clusters in our analysis did not differ and thus likely did not bias individual clusters' outcomes, plus given the substantial discrepancy between the significantly increased odds to achieve CR and the subsequent meager OS of cluster A, a closer monitoring of patients fitting that cluster seems warranted and potentially, allogeneic HCT may be an option for these patients to consolidate the initial treatment response. With respect to high-risk disease, cluster B showed dismal rates of treatment response

and substantially decreased OS. In a proof-of-concept fashion, B showed the highest rate of established markers of poor outcomes such as complex karyotypes, −5/del(5q) as well as mutated *TP53*, *RUNX1*, *ASXL1*[4,35–40] paired with an absence of established good risk markers. Considering ELN2017[4] allocation, patients in B majorly either belonged to the ELN2017 intermediate or high-risk groups while a small proportion were labeled ELN2017 favorable. In contrast, cluster D showed remarkably favorable outcomes with high rates of CR and substantially prolonged survival and was mainly comprised of patients bearing normal karyotypes as well as mutations of *CEBPA*, dominated by biallelic mutations as well as single mutations in the bZIP and TAD domain, followed by mutated *GATA2* and *TET2* in the absence of high-risk markers corresponding to a majority of patients being allocated to the ELN2017 favorable group followed by ELN2017 intermediate group. Biallelic mutations of *CEBPA* have been demonstrated to be associated with improved outcomes[41,42], and recently a differing prognostic impact of mutational sites of *CEBPA*-TAD and -bZIP has been reported[23]. Mutated *GATA2* has been reported to be associated with biallelic mutations of *CEBPA*[23,43] and improved outcome[44], however whether the latter is due to *GATA2* mutation status alone or related to the frequent co-mutational spectrum of *GATA2* is unclear. Interestingly, roughly a third of patients in D also harbored mutations in *TET2*, which have been reported to be associated with poor outcomes[45–47]. Lastly, patients in cluster C showed intermediary outcomes compared to the other clusters and were allocated in equal parts of ELN2017 favorable and intermediate risk categories with a minority displaying ELN2017 adverse features. Notably, C harbored the highest proportions of t(8;21) and inv(16) or t(16;16), both have been previously associated with improved outcomes[4,48,49], however, were not allocated to the 'good-risk' cluster D by unsupervised learning in our analysis.

A key limitation in the transferability of any unsupervised learning task is the data set used for cluster generation. It has to be acknowledged that both our internal data set for cluster generation and our external data set for validation stem from a multicenter collective of central European university centers and a patient cohort that was treated in multiple anthracycline-based intensive regimens within previously reported clinical trials. Such a patient cohort obviously falls short to acknowledge ethnical differences in disease biology, differences in supportive treatment between healthcare systems and overall differences between patients that are eligible for intensive therapy within a clinical trial and those who are not. We want to point out that study results are not set in stone but our dynamic approach easily enables the pooling of patient data with other cohorts from different backgrounds in order to generate a more holistic picture of AML's molecular and clinical landscape than rigid hypothesis-driven models can that need to be updated manually over the course of years. However, for this purpose it is necessary to actually pool data from different international sources and run cluster analysis on them. By incorporating a final supervised learning step into our pipeline, future research may also seek to validate cluster assignments in a prospective fashion by assigning patients to clusters pre-treatment and compare predicted cluster results to actual patient outcomes.

As pointed out, low-scale hypothesis-driven models likely underestimate the complexity of AML biology. While our model is not intended or validated to challenge well-established risk assessment tools, we argue that data-driven models can potentially leverage a multitude of available clinical and genetic information for a more nuanced and personalized approach to AML therapy. For instance, Gerstung et al.[50] demonstrated that the use of knowledge banks incorporating predictions of treatment response and survival for a variety of treatments may allow for

individual treatment regimens reducing the need for allogeneic HCT in 20–25% of patients while maintaining overall survival rates. In that sense, unsupervised cluster analysis may aid in unveiling differences and commonalities in disease biology that are of prognostic impact as, for example, demonstrated by Bullinger et al.[51,52]. Additionally, Awada et al.[53] recently used Bayesian latent class clustering to group AML patients based on gene sequencing data and similarly found 4 distinct genomic clusters with impact on patient outcomes while challenging the conventional dichotomy between de novo and sAML.

From a methodological perspective, a common shortcoming in biomedical machine learning is the reduction of a given use-case to one single algorithm without evaluation of alternative algorithms in comparison. Further, in contrast to conventional classification tasks (in supervised learning), clustering of patient data without ground truth labels is inherently difficult to evaluate. In supervised learning an algorithm is trained with labeled samples and a well-defined ground truth and is tested by predicting labels to previously unseen samples while being measured on how well the predicted labels fit the ground truth. Contrastingly in unsupervised learning, no labels exist and the ground truth, if one exists at all, is often difficult to point to. Hence, it is of utmost importance to inspect clustering results in their specific contextual domain, respective research question and, most importantly, evaluate them by information gain. By incorporating a multitude of different algorithm combinations into a single model and subsequently meta-clustering results with minimal feature engineering, our approach acknowledges differences between algorithms that usually cannot be determined a priori for a given data set as no 'one-size fits all' solution exists in unsupervised learning and results may vary based on data transformation and clustering methods. Despite our analysis on multi-center data stemming from over 50 centers, the retrospective nature of this model is still a limitation and prospective validation is nonetheless warranted. Additionally, our approach is limited by excluding sparse features to avoid the curse-of-dimensionality[26]: An increased number of variables, i. e. higher model dimensionality, inflates the model space rapidly which leads to a scarcity of available data in the model. In order to avoid the consequential model instability and ensure reliable and interpretable results, we excluded parameters that were only present in <1% of patients at the pre-processing state and the remaining parameters were mapped to a lower dimensional space. Potentially, this may lead to a loss of information, which however can be tackled by including more data, i. e. more patients. Hence, a larger sample of patients may allow for rare features, combinatorial features (i.e. unmutated *NPM1* + *FLT3*-ITD$_{high}$) or higher dimensional mapping to be included without sacrificing model stability while potentially increasing the information gain. With the availability of a much larger patient cohort, e. g. through collaborative data collection efforts in hematology such as HARMONY[54], clustering could be extended to methods such as deep learning[55], potentially unveiling more nuanced differences between patient groups. Additionally, all patients in our cohort received intensive chemotherapy, and – with the exception of patients from the SOR-AML trial[17] which showed no impact on long-term overall survival for the addition of sorafenib[56]—did not receive molecular targeted therapy. With the advent of targeted therapy in AML treatment, more personalized treatment regimens will have impact on individual patient outcomes[57,58]. Hence, a comprehensive evaluation of a wide range of potentially impactful molecular lesions for a given patient seems reasonable and a dynamic data-driven approach as ours to delineate patient groups according to their risk profiles possibly also stratifying for different treatment modalities in the future is likely more suitable to keep pace with the rapid advancements in targeted therapies than rigid hypothesis-driven models. Potentially, our model is transferable to other cancer entities with multi-modal data. Hence, the code used for the purpose of this work is publicly available.

In summary, we here provide a data-driven approach combining a variety of unsupervised learning algorithms and meta-clustering to identify distinct patient clusters that differ significantly in clinically meaningful endpoints like achievement of CR and survival based on heterogenous clinical, laboratory and genetic data. We evaluated our model on patient data from a multinational cohort from 59 centers and externally validated clustering results. Our model provides further insight into the complex biology and clinical presentation of AML beyond ELN2017 recommendations and warrants the incorporation of data-driven models utilizing large biomedical data sets for a more accurate stratification of AML patients in order to improve patient care.

## Data availability
Data that was used for initial clustering in this study stems from previously reported multi-center trials (AML96[14], AML2003[15], AML60 + [16], and SORAML[17]) of the SAL. Data that was used for external validation stems from previously reported trials of the AMLCG (AMLCG-1999 and AMLCG2008)[19,20]. Further information on patients enrolled in these trials can be obtained from the individual references. Data is available from the corresponding author upon reasonable request. Full public access is currently not possible due to ongoing retrospective studies of the respective study alliances with currently unpublished results. The underlying data for the figures generated for the purpose of this study can be found in Supplementary Data 7 (Fig. 3), Supplementary Data 8 (Fig. 4), Supplementary Data 9–12 (Fig. 5).

## Code availability
The code[59] generated for the purpose of this study is publicly available at https://zenodo.org/badge/latestdoi/629506656.

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

## Acknowledgements

We thank all contributing physicians, laboratories and nurses associated with the German Study Alliance Leukemia and especially participating patients for their valuable contributions. J.-N.E. is grateful for research support from a scholarship of the Mildred-Scheel-Nachwuchszentrum (German Cancer Aid).

## Author contributions

J.-N.E., K.W., M.B., C.T. and J.M.M. designed the study. J.-N.E., C.R., J.-A.G., F.K., F.S., U.P., K.M., K.S., U.K., J.B., D.G., C.Sa., B.W., T.H., W.H., C.M.-T., H.S., C.D.B., K.S.-E., M.K., S.W.K., M. Häne., W.E.B., C.Sch., M. Hano., J.M., J.S., M.B., C.T., J.M.M. provided patient samples. S.S., J.-A.G. and C.T. performed molecular analysis. J.-N.E., P.H. and K.W. developed and implemented the machine learning framework. J.-N.E. performed statistical analysis, generated visualizations and wrote the draft. All authors had access to all of the data, analyzed the data, provided critical scientific insights and revised the draft. All authors agreed to the final version of the manuscript and the decision to submit it for publication.

## Funding

## Competing interests

The authors declare no competing interests.

## Additional information

[1]Department of Internal Medicine I, University Hospital Carl Gustav Carus, Dresden, Germany. [2]Else Kröner Fresenius Center for Digital Health, Technical University Dresden, Dresden, Germany. [3]Medical Clinic and Policlinic I Hematology and Cell Therapy, University Hospital, Leipzig, Germany. [4]Department of Software and Multimedia Technology, Technical University Dresden, Dresden, Germany. [5]Laboratory for Leukemia Diagnostics, Department of Medicine III, University Hospital, LMU Munich, Munich, Germany. [6]Department of Medicine III, Hospital Leverkusen, Leverkusen, Germany. [7]Hospital Barmherzige Brueder Regensburg, Regensburg, Germany. [8]Institute for Biostatistics and Clinical Research, University Muenster, Muenster, Germany. [9]Department of Hematology, Oncology and Tumor Immunology, Charité, Berlin, Germany. [10]Department of Medicine V, University Hospital Heidelberg, Heidelberg, Germany. [11]German Consortium for Translational Cancer Research DKFZ, Heidelberg, Germany. [12]Department of Medicine 2, Hematology and Oncology, Goethe University Frankfurt, Frankfurt, Germany. [13]Department of Hematology and Oncology, University Hospital Schleswig Holstein, Kiel, Germany. [14]Department of Internal Medicine 5, University Hospital Nuremberg, Nuremberg, Germany. [15]Department of Hematology, Oncology and Palliative Care, Robert-Bosch Hospital, Stuttgart, Germany. [16]Department of Internal Medicine 5, University Hospital Erlangen, Erlangen, Germany. [17]Department of Internal Medicine 3, Klinikum Chemnitz GmbH, Chemnitz, Germany. [18]Department of Internal Medicine A, University Hospital Muenster, Muenster, Germany. [19]Department of Internal Medicine, Hematology and Oncology, Masaryk University Hospital, Brno, Czech Republic. [20]Department of Hematology and Stem Cell Transplantation, University Hospital Essen, Essen, Germany. [21]National Center for Tumor Diseases (NCT), Dresden, Germany. ✉email: jan-niklas.eckardt@uniklinikum-dresden.de

