## [Peer Review File · Communications Medicine]

Reviewers' comments:

Reviewer #1 (Remarks to the Author):

The study titled “Unsupervised Meta-Clustering Identifies Risk Clusters in Acute Myeloid Leukemia Based on Clinical and Genetic Profiles” proposes novel unsupervised clusters via meta-clustering incorporating dimension reduction and clustering in a combinatorial fashion. Although the study is novel in the use of meta-clustering, key aspects are left out regarding the utilized methods and more rigorous comparisons are required.

Major Comments

1. I think that manuscript should benefit from clearly stating the goals/objectives of the clustering: was it to mechanistically establish convergent subtypes or to better predict survival?
2. The fraction of sAML appears rather low suggesting that selection of patients may be biased e.g. through participation criteria in the listed clinical trials. Would the results of clustering be changed, should e.g., more sAML cases be included. To that end, how age of the patients affects the scoring: e.g., patients with sAML may be older and receive less aggressive therapy? Would the result hold only for sAML? If not what would be the conclusions for molecular impact of lesions: I assume that sAML had less NPM1 and FIt3 or t 8/21 (examples) and thus sAML if enriched may cluster differently or different clusters would have been computed.
3. Is there a validation cohort that would, using the same methodology, produce similar clustering results and similar survivals.
4. I cannot find transplant information sorry—Please, state that HSCT recipients were excluded censored etc for KM analyses or were separately analyzed.
5. Please better described key minimal features allowing for the recapitulation of the clusters: are all 61 parameters needed?
6. I cannot easily find the 61 parameters used for clustering I assume that these were only molecular parameters (mutations and cytogenetic features). The description is not very clear as the authors start the results with a final output.
7. It seems that the resolution of cluster by survival according to KM was a criterion (sanity check) at some point. This however may impair the functional relationships with clusters: was the best resolution of survival a criterion? As such one would think that e.g. short survival may be function of various not disease related factors and treatment. E.g., patient who died due to sepsis or car accident early may have totally different genetics..... or patients with distinct genetic profiles may have similar survival. This needs to be clarified and discussed. What was the overall aim of this analysis.
8. The quality of prognostic prediction DFS, OS etc... could be assessed e.g. by C scores estimated vs actual survival since the analysis is retrospective
9. The treatment modalities in each cluster should be shown. How many patient received what?
10. The authors describe a combinatorial workflow of 605 unsupervised clustering methods. However, important details regarding the selection of the methods, optimization of parameters and aggregation of clustering results are missing. For instance, the authors utilize both the PCA and SVD which are indistinguishable in practice and result in the same projections. Similarly, AE model is not described, nor are the parameters involved (number of layers/units of layers etc.).

11. The authors combine multiple transformation/dimension reduction methods with clustering. However, not all of the transformations described are suitable for different types of inputs. For instance, principal component based methods assume the signal to follow a Gaussian distribution for which categorical variables/one-hot encoded variables are not appropriate. Similarly, not all of the transformations are applicable for the utilized clustering methods. Non-linear transformations are better suited for density based clustering (DBSCAN) vice versa hence an explicit weighting might be required during aggregation which is not described as well.

12. The authors describe no validation approaches utilized for the identified clusters. One straightforward approach would be to do a k-fold cross-validation at least with a subset of the combinatorial workflow.

13. Finally, I find the conclusions modest, ie: essentially AML is more complex than ELN can appreciate". A list of concrete new results findings should be provided.

Minor Comments

1. Regarding the Figure 3, KM curves, confidence bands for individual curves and the number of samples at risk should be given.

2. It would be informative if overlap of identified clusters (e.g. adjusted rand index) across different workflows were shown in order to quantify the utility of using multiple workflows better.

3. Discussion is very dense

Reviewer #2 (Remarks to the Author):

I am happy to sign this report to the authors. I am David Westhead from the University of Leeds.

I think this is an interesting piece of work, but I would raise some issues that I think need attention.

1. First and foremost, I think that to have substantial influence on the field this work needs to be pushed a bit further. I would like to see some effort at validation, ideally in an independent cohort of patients, rather than just a clustering of a single cohort. In the absence of a suitable independent cohort it would in this context have been useful to hold out a validation/test from the data set used. There is always a risk that biases in any one cohort lead to methods that don't generalise well in different cohorts, and also that data analysis decisions introduce cohort dependent bias in method and parameter selection. Since this data is already a union of cohorts of patients from different sources you might consider whether using just one of those sources as test/validation would be a good option?

You comment in the discussion that prospective evaluation would be needed, and for this you will need a method to assign a new patient to one of your clusters, i.e. a classifier. Developing this for the final clustering solutions in the training data set would then give a way

of assigning cluster membership in the test/validation data. And with that you could check the expected associations of cluster membership with response and survival in the validation/test data. This would strengthen the paper substantially.

2. Your treatment of missing data is a concern. Exclusion of variables present of <1% of the patients is a relatively weak condition and leaves other variables in with significant missingness. The approach of just replacing with the median needs to be considered. Missingness may not be completely at random, and there is a danger that with naive treatment you can inadvertently code other information about the patients. For instance, if doctors order tests on the basis of some prognostic information then missingness of these tests can contain prognostic information. Machine learning methods can discover this and learn patterns that are not useful, and it could contribute to the observed survival/response differences between these clusters. I would recommend a more careful treatment of missing data, considering these effects and evaluating as well more sophisticated methods of multiple imputation, as well as complete case analysis if feasible.

3. In considering differences in variables between your clusters I would refrain from attaching p values to variables that are used in clustering. These are not really meaningful: by clustering on a variable and then comparing patients in different clusters you are selecting groups of patients that are already expected to differ on the clustering variable. That means that the p values are not meaningful in the usual sense, and you could argue that they really only prove that your clustering is working.

4. The meta clustering algorithm needs detailed explanation for reproducibility. It is important to understand how the eventual solution of 4 clusters is arrived at and in what sense it is optimal.

5. Further to point 4, the ideal solution to reproducibility is to provide a script (or perhaps a Jupyter notebook or similar) that does all the analysis from raw data input to output figures and tables for the paper. I know this is hard work, but I think it is something to which we should aspire, and it saves work for everyone who wants to use or extend the work in the future.

6. Finally, the presentation of methods after results does mean that in places you need to give just a bit more information in the results section. Some statements are hard to interpret without reading ahead to methods, e.g. 'aggregated the results of 605 different combinations of target dimensionalities' needs just a bit more local explanation to be meaningful to read. I can probably find other examples.

Point-by-Point Response

We want to thank the editor for the opportunity to revise our manuscript for resubmission and the anonymous reviewers for their insightful comments on our work.

We substantially revised our manuscript according to the reviewers' comments and provide a point-by-point response below. The revised sections in the manuscript are highlighted using Microsoft Word's 'track changes' function.

We want to stress one fundamental aspect before going into detail: Our work has previously been based solely on an unsupervised learning approach. Most commonly in the translational research community at the intersection of AI and medicine, we find technologies stemming from the supervised learning domain. Research questions and data sets are usually pre-processed and formulated in a way that by training on a certain cohort X a pattern is learned and validated on a certain (external) cohort Y. For example, this is increasingly prevalent in the literature with regard to imaging techniques e. g. from histopathology and radiology. By learning from well-labeled data, supervised learning can easily be transferred to a number of external cohorts where the same model can be evaluated on whether or not it can perform on said cohort by simply comparing the number of accurately predicted labels with the ground truth. Unsupervised learning is fundamentally different. In unsupervised learning, there are no labels and hence, there is essentially no prediction as the ground truth is not known. For example, in supervised learning we may know whether a chest X-ray depicts pneumonia or is a healthy control and we may train a supervised model to predict pneumonia and judge its performance based on its hit-rate. In unsupervised learning, the entire problem is different as this method aims at clustering data according to its features rather than predicting any label. Therefore, transferring any unsupervised method to another data set is not a trivial task as in supervised learning since instead of learning to predict a pattern, unsupervised learning is rather used for sorting data and, obviously, different data sets may be sorted differently. From a technological perspective, this posed a substantial challenge in revising our manuscript as there are no 'off-the-shelf'-techniques to externally validate an unsupervised model as there are in supervised learning. Further, in order to obtain a meaningful abstraction of the model, we required a dataset with a large overlap in terms of patient features as the introduction of novel features or the absence of features that are present in our data set may have fundamentally distorted the output shape of the final model. Nevertheless, we fully understand and support the reviewers' demand for external validation. In order to provide the scientific community with a meaningful tool, we would need to also provide a means of incorporating new data into the model.

Hence, we obtained an external validation cohort that largely overlaps in captured features with our original data set. The external cohort consists of 664 AML patients that also received intensive anthracycline-based regimens and were treated within the scope of previous clinical trials from the AMLCG, a large multicenter cluster of university centers specialized in the treatment of hematological malignancies from southern Germany and Austria.

Further, we converted the external validation into a supervised learning problem. We used the cluster assignments that were given as an output by our unsupervised model as labels for the patients from our cohort. Then, we trained a supervised model in order to abstract features from the original patients that can be used to provide cluster assignment labels to novel patients from the external cohort. This method not only conserves the original unsupervised approach and its output, but fundamentally adds a component of transferability to other data sets, provides reproducibility and serves as a proof-of-concept from a technological point-of-view for transferring the output of a clustering algorithm to a supervised model.

We hope the revised version of our manuscript is to the editor's and the reviewers' full satisfaction. Please find the responses to your comments and revisions from the manuscript text down below.

Kind regards,

the authors

Reviewers' comments:

Reviewer #1 (Remarks to the Author):

The study titled "Unsupervised Meta-Clustering Identifies Risk Clusters in Acute Myeloid Leukemia Based on Clinical and Genetic Profiles" proposes novel unsupervised clusters via meta-clustering incorporating dimension reduction and clustering in a combinatorial fashion. Although the study is novel in the use of meta-clustering, key aspects are left out regarding the utilized methods and more rigorous comparisons are required.

Major Comments

1. I think that manuscript should benefit from clearly stating the goals/objectives of the clustering: was it to mechanistically establish convergent subtypes or to better predict survival?

We agree with the reviewer that the previous version of the manuscript did not clearly state its goal. As we highlight above, in contrast to supervised learning, unsupervised learning cannot provide any meaningful prediction but rather sorts patients into clusters. Sorted clusters, however, can be utilized to answer a variety of clinical questions from overlapping disease biology to patient outcome. Importantly, outcome can only be evaluated on a cluster level rather than a patient level.

We substantially revised the discussion to not only make the goal of our study, but also the potential lessons for the clinical readership clearer. In essence, we show that conventional checklist-like models such as the ELN recommendations fail to acknowledge disease complexity as they treat patient features basically like a decision tree: Feature A is present in a patient, feature A is associated with adverse risk, therefore, patient is

categorized as adverse risk. This fails to acknowledge that feature A may act in a molecular context with other features that may alter its behavior and therefore the patient's risk profile. Further, such expert-opinion-based solutions fail to capture the rapid pace of scientific research findings in AML (and other entities') biology. We believe a rather data-driven approach that can be re-iterated on large multi-center data may be more suitable to keep up with the advancements in the field. Our model is intended to serve as a first step in this direction. Future development necessarily encompasses the incorporation of AML patients from other countries and, importantly, other ethnicities than middle European patients and, of course, other treatment regimens than intensive anthracycline-based therapy. Given our restructuring of the pipeline to incorporate a final supervised learning step for external validation, a follow-up project can encompass a prospective validation, although this requires more participating centers and obviously is not within the scope of the current study.

2. The fraction of sAML appears rather low suggesting that selection of patients may be biased e.g. through participation criteria in the listed clinical trials. Would the results of clustering be changed, should e.g., more sAML cases be included. To that end, how age of the patients affects the scoring: e.g., patients with sAML may be older and receive less aggressive therapy? Would the result hold only for sAML? If not what would be the conclusions for molecular impact of lesions: I assume that sAML had less NPM1 and Flt3 or t 8/21 (examples) and thus sAML if enriched may cluster differently or different clusters would have been computed.

We agree that the proportion of saML is rather low compared to other studies, although still at the lower boundary of what other large cohorts have reported (e. g. Leone et al. Haematologica, 1999).

First of all, we need to clarify that there are no individual weights for individual features within the model. The way this essentially works is that for n-features an n-dimensional coordinate system is computed with each axis representing the feature distribution for each given feature. These features are normalized to the z-score to make different scales comparable. Since computing in $n > 200$ dimensions is computationally rather impossible as long as one is not using a large cluster of high-performance computers, we need to downscale features and transform the data to a lower dimension that is computationally more feasible to operate in. That means, that the shape of the data highly depends on features used and their distribution within the cohort. As you correctly point out, altering the number of patients with secondary AML will inevitably alter the final shape of the data and hence the output. What the output in such a case would be, we cannot say without conducting the experiment. This ties in with our response to your first comment: What is needed is a large collaborative effort from multiple centers and countries in order to obtain a data set that is more in line with the overall (worldwide) AML patient population rather than a central European one. Hence, we cannot really answer what the

impact for a sAML-enriched cohort would be with regard to cluster assignment and molecular make-up of the cohorts without testing our approach for a sAML-enriched cohort. However, as our code is freely available, anyone with such a cohort may run it after the necessary pre-processing steps to tailor the pipeline to the respective data set and see what the outputs are.

As you correctly point out, this represents a limitation of the study that is not acknowledged sufficiently in the previous version of the manuscript. We added this limitation to the discussion:

“A key limitation in the transferability of any unsupervised learning task is the data set used for cluster generation. It has to be acknowledged that both our internal data set for cluster generation and our external data set for validation stem from a multicenter collective of central European centers and a patient cohort that was treated in multiple anthracycline-based intensive regimens within previously reported clinical trials. Such a patient cohort obviously falls short to acknowledge ethnical differences in disease biology, differences in supportive treatment between healthcare systems and overall differences between patients that are eligible for intensive therapy within a clinical trial and those who are not. We want to point out that study results are not set in stone but our dynamic approach easily enables the pooling of patient data with other cohorts from different backgrounds in order to generate a more holistic picture of AML’s molecular and clinical landscape than rigid hypothesis-driven models can that need to be updated manually over the course of years. However, for this purpose it is necessary to actually pool data from different international sources and run cluster analysis on them. By incorporating a final supervised learning step into our pipeline, future research may also seek to validate cluster assignments in a prospective fashion by assigning patients to clusters pre-treatment and compare predicted cluster results to actual patient outcomes.”

lines 299-313

3. Is there a validation cohort that would, using the same methodology, produce similar clustering results and similar survivals.

As pointed out above, for unsupervised learning tasks the overlap of available features is essential when considering transferability to another cohort. Therefore, it is not an easy undertaking to find a patient cohort that essentially represents the same data distribution as the initial cohort. After an intensive search, we obtained a large cohort of 664 intensively treated AML patients from the AML Cooperative Group, a large multi-center study group encompassing university centers in southern Germany and Austria.

Simply re-running the clustering on this cohort would not provide us with any meaningful information abstraction with regard to our original clustering results. As we want our results but more importantly our method to be applied in a clinical context, we needed to devise a method to connect the initial clustering to the new cohort. As highlighted above, there are no ‘off-the-shelf’-solutions for this in unsupervised learning.

This is diametrically different from supervised learning where a learned pattern can simply be transferred to another data set by predicting the same labels on new data and see how accurate the model is. In unsupervised learning, however, there is no ground truth and thus no labels. From the AI perspective, this is a technological challenge. We reformulated this problem in the following way: The output of our meta-clustering provides a final cluster assignment for each patient individually that connects the patient to other patients within the same cluster. This means that fundamentally there are features that determine cluster assignment. These features can potentially be learned in a supervised learning task once the unsupervised labels (=cluster assignments) are established. Hence, after the meta-clustering we added a step for supervised learning in order to predict cluster assignments on an external cohort (see updated Figure 1). This allows us to conserve the original cluster assignments and add a layer of supervised learning to incorporate any number of external patients. Potentially, this method can be re-iterated indeterminately: A given set of n patients can be used for cluster generation and another set of m patients can be added for external validation. This may help to validate clustering results and also enables a potential prospective validation as any number of patients may be added at any point in time to predict cluster assignment.

Updated Figure 5:

Figure 5. Step-wise workflow of unsupervised learning. After pre-processing of multimodal patient data (1), unsupervised learning (2) was performed with different combinations between target dimensionality (2 – 6, n=5), data transformation (n=11) and

unsupervised clustering algorithms (n=11). The individual outputs of each algorithm combination were gathered and used as input for meta-clustering (3) to find patient clusters that in themselves are maximally homogenous while at the same time differ maximally from other patient clusters. Thus, final clusters are identified and individual features of patients within these clusters become available for further analysis. In the next step, the previous cluster assignments can be treated as labels for supervised learning (4). After training and testing on the original cohort, the highest performing classifier is being selected for assigning cluster labels to an external validation cohort.

Updated Methods:

External validation as a supervised learning task

Unsupervised learning inherently only sorts samples into clusters. Therefore, any addition of new data will lead to a completely new sorting. Ideally, original clusters would be retained, however, this depends on the distribution of the newly introduced data. In order to externally validate our model, a mere addition of data is therefore insufficient. Unsupervised learning does not consider labels as there is no evident ground truth, but rather commonalities and differences between samples are at the center of the analysis. Contrastingly in supervised learning, labels can be used to learn features that distinguish a certain set of specimens (e. g. shape and color can be learned to distinguish apples and bananas). Hence in our set-up cluster assignment can be viewed as labels that are determined by patient features. Therefore, in order to include new data into the model, the set-up can be modified to a supervised learning task as soon as one set of final cluster assignment labels has been generated previously. Potentially, this process can be iterated ad infinitum: Use unsupervised clustering for cohort A, predict cluster assignments based on A's features for cohort B. It could also be modified in order to alter the original cluster results: Use unsupervised clustering for cohort A + B, predict cluster assignments based on A and B's features for cohort C etc. For this design to function properly, the overlap in available features between cohort A and B has to be high (ideally fully matching). We obtained an external cohort of 664 intensively treated AML patients from previous clinical trials as described above. Cluster assignment labels were learned using supervised learning on the original cohort and subsequently, cluster assignment was predicted on the external cohort in order to sort them within the existing clusters. Pre-processing for the external data did not differ from our original cohort as described above. Again, there is no universally 'best' algorithm for such a supervised learning problem. Therefore, different algorithms have to be evaluated based on their individual performance for a given task. To do this, we used a train-test-split on the original cohort of 80:20 and evaluated four different supervised algorithms: naïve Bayes, gradient boosting, random forest and logistic regression. Algorithm performance was evaluated using AUROC, precision, recall, and F1-score. The overall best performing supervised algorithm was selected in order to assign cluster labels to the external validation cohort. Since there is no ground truth for cluster labels with regard to the external cohort, only test set performance on the original cohort can be reported. Based on these cluster

assignments, survival analysis was performed on the external cohort and compared to our original cohort.

Lines 544 - 571

Updated Results:

Cluster outcomes are preserved in an external multicenter validation cohort

In stark contrast to supervised learning, external validation of an unsupervised learning task is much harder to accomplish since there is no obvious ground truth an algorithm's performance can be compared against. Furthermore, there is no 'off-the-shelf'-solution to incorporate new data into a previously established cluster without altering the cluster itself. In order to retain previously generated clusters and at the same time add new external data, we transformed the task into a supervised one by using the cluster assignments as learnable labels. We then trained four different supervised algorithms to predict cluster assignments on the original cohort with a 80:20 train-test-split in order to identify the most suitable algorithm for the task. Individual model performance is displayed in Table S2. With a test set AUROC of 0.99, logistic regression was the highest performing algorithm in cluster assignment. Hence, logistic regression was used to assign cluster labels to an external cohort. This cohort was comprised of 664 intensively treated AML patients from previous trials of the German-Austrian multicenter AML Cooperative Group (AMLCG). In comparison to the original cohort, cluster sizes for each of the four clusters were similar. In general, the directionality of effects, i. e. cluster B showed overall a high risk-profile while cluster D had more favorable outcomes with clusters A and C in between, was retained. As in the original cohort, patients assigned to cluster A were significantly more likely to achieve CR (OR = 2.08). Patients assigned to B again had dismal outcomes as they were significantly less likely to achieve CR (OR = 0.33) and had a significantly decreased RFS and OS as in the original cohort with a HR of 1.68 and 1.87, respectively. C, again, showed intermediate outcomes with regard to survival times, however, in contrast to the original cohort none of these were statistically significant. Lastly, patients assigned to D had the most favorable outcomes with significantly increased RFS and OS (HR = 0.47 and 0.64, respectively) as in the original cohort, while CR rate did not differ in contrast to the original cohort. Detailed information on outcomes in comparison to the original cohort can be taken from Table 3 and Fig. 3D-E. Information on EFS was not available for the external validation cohort.

Lines 202-223

SAL multicenter investigational cohort				
Cluster A	Cluster B	Cluster C	Cluster D	all patients

CR after induction therapy, n (%)	335 (79.0)	140 (54.7)	390 (72.8)	143 (85.6)	1008 (72.9)
OR	1.60	0.36	0.99	2.42	
[95%-CI]	[1.22 - 2.10]	[0.27 - 0.48]	[0.78 - 1.26]	[1.54 - 3.79]	
adj. p-value	0.006	<0.001	0.934	<0.001	
median EFS, months (IQR)	8.3 (1.7 - 48.3)	1.9 (0.4 - 9.6)	9.1 (1.2 - n. r.)	11.2 (3.6 - n. r.)	7.1 (1.2 - 35.4)
HR	0.96	1.84	0.83	0.74	
[95%-CI]	[0.85 - 1.10]	[1.59 - 2.14]	[0.73 - 0.94]	[0.61 - 0.90]	
adj. p-value	0.700	<0.001	0.009	0.007	
median RFS, months (IQR)	14.8 (5.1 - n.r.)	11.9 (4.1 - 108.3)	30.7 (8.1 - n.r.)	23.9 (5.4 - n.r.)	17.5 (5.6 - 71.4)
HR	1.25	1.39	0.73	0.88	
[95%-CI]	[1.06 - 1.47]	[1.12 - 1.72]	[0.62 - 0.86]	[0.70 - 1.11]	
adj. p-value	0.011	0.008	<0.001	0.400	
median OS, months (IQR)	17.1 (6.2 - n.r.)	10.5 (4.2 - 33.2)	25.8 (7.1 - n.r.)	54.0 (7.6 - n.r.)	17.2 (6.0 - 59.7)
HR	1.05	1.69	0.82	0.68	
[95%-CI]	[0.91 - 1.20]	[1.44 - 1.97]	[0.71 - 0.94]	[0.55 - 0.85]	
adj. p-value	0.68	<0.001	0.007	0.001	

AMLCG multicenter validation cohort

	Cluster A	Cluster B	Cluster C	Cluster D	all patients
CR after induction therapy, n (%)	172 (77.0)	50 (45.0)	189 (66.1)	34 (77.3)	445 (67.0)
OR	2.08	0.33	0.93	1.73	

[95%-CI]	[1.44 – 3.00]	[0.22 – 0.50]	[0.67 – 1.29]	[0.84 – 3.57]	
adj. p-value	<0.001	<0.001	0.656	0.139	
median EFS, months (IQR)					
HR	-	-	-	-	-
[95%-CI]					
adj. p-value					
median RFS, months (IQR)	15.7 (5.7 - 55.5)	7.0 (4.1 – 46.4)	20.4 (7.5 – 71.6)	n.r (8.5 - 104.2)	16.3 (6.4 - 61.4)
HR	1.25	1.68	0.81	0.47	
[95%-CI]	[0.97 - 1.62]	[1.14 - 2.48]	[0.62 - 1.04]	[0.26 - 0.87]	
adj. p-value	0.088	0.009	0.103	0.015	
median OS, months (IQR)	20.8 (7.5 - 70.0)	7.7 (7.7 - 24.3)	20.6 (5.6 – 73.2)	70.9 (10.8 – 100.8)	17.4 (14.4 - 21.6)
HR	0.85	1.87	0.92	0.64	
[95%-CI]	[0.70 - 1.03]	[1.49 - 2.35]	[0.76 - 1.11]	[0.42 - 0.96]	
adj. p-value	0.110	<0.001	0.382	0.031	

Table 3. Differences in outcome according to clusters identified by unsupervised learning. Odds ratios (OR) for achievement of complete remission (CR) were obtained using univariable logistic regression. Hazard ratios (HR) for event-free survival (EFS), relapse-free survival (RFS) and overall survival (OS) were obtained using univariable Cox proportional hazard models. 95%-confidence intervals (95%-CI) and adjusted p-values obtained with the Benjamini-Hochberg method are reported for all calculations as well as the interquartile range (IQR) for survival times. Other abbreviations: n: number; n.r.: not reached.

Updated Figure 3:

Figure 3. Differences in survival times between clusters. Survival times with regard to event-free (EFS), relapse-free (RFS) and overall survival (OS) were compared using the Kaplan-Meier-method and log-rank test. Results for the original cohort are shown in panels A to C. These differed significantly between the four clusters: While cluster D showed relatively favorable outcomes followed by cluster C, contrastingly, B was

characterized by poor long-term survival followed by cluster A. This is most evident regarding EFS and OS while in RFS, both A and B as well as C and D were largely overlapping. For the external validation cohort, no information regarding EFS were available. With respect to RFS and OS, clusters again showed significantly different outcomes with D bearing the most favorable and B being the least favorable outcome. Numbers at risk are shown for the respective time points (0, 50, 100, 150 months). Colored bands represent 95%-confidence intervals.

4. I cannot find transplant information sorry—Please, state that HSCT recipients were excluded censored etc for KM analyses or were separately analyzed.

Information on HSCT recipients is provided within Table 1:

	Cluster A	Cluster B	Cluster C	Cluster D	adj. p-value	all patients
HSCT in first remission, n (%)	69 (16.3)	33 (12.9)	86 (16.0)	39 (23.4)	0,058	227 (16.4)
HSCT as salvage therapy, n (%)	68 (16.0)	34 (13.3)	88 (16.4)	28 (16.8)	0,675	218 (15.8)

We understand the reviewer’s concern that transplantation may substantially alter patient outcome and thus could be a potential confounder in analyzing survival times between clusters. We want to stress that features used for clustering explicitly excluded outcome variables such as achievement of CR and survival times (EFS, RFS, and OS) plus HSCT status, i. e. HSCT and outcome status were not features used for cluster generation! We understand that this is a bit cryptic in the previous version of the document and address this issue by providing a list of features used for clustering (see response to comment 6).

Further, Table 1 evidently shows that the fractions of patients who ultimately received HSCT did not differ significantly between any of the four clusters and thus should not influence outcome of one (or any number) of the clusters. Hence, we can likely assume that HSCT is not a confounder in subsequent cluster analysis.

As this information is admittedly a bit hidden in Table 1, we added a sentence to highlight this in the results as well:

“Allogeneic hematopoietic stem cell transplantation (HSCT) status was explicitly not used as a feature for cluster generation as it may have substantially biased results with regard to survival times. The rate of HSCT either upfront or as salvage treatment did not differ

significantly across clusters (adj. p=0.058 and adj. p=0.675, respectively). Therefore, any potential effect of HSCT on cluster outcome appeared to be equally distributed among clusters rather than distorting outcomes for any cluster in particular.”

Lines 158-163

5. Please better described key minimal features allowing for the recapitulation of the clusters: are all 61 parameters needed?

6. I cannot easily find the 61 parameters used for clustering I assume that these were only molecular parameters (mutations and cytogenetic features). The description is not very clear as the authors start the results with a final output.

We totally agree with the reviewer and apologize for the inconvenience. We are somewhat restricted by editorial guidelines in the number of tables and figures we are allowed to display. The information on what variables exactly is used for cluster generation is unfortunately not easy to find. In the previous version of the manuscript, this has been shown in the Supplements in Table S1 (previous version). However, as the reviewer points out, it is much more appropriate to provide the reader with a comprehensive list in the main document. Therefore, we added the table to the main document as the new Table 1 in the revised version of our manuscript. We hope the editor approves of this decision even if this constitutes an additional display item. We believe it is essential for the reader to make sense of how cluster generation was performed to have this item in the main document. We would prefer not to transfer neither table 2 (baseline characteristics of the clusters) nor table 3 (outcome of the clusters) to the extended data or supplements as we also believe this information is of high relevance to the reader. We hope the editor approves.

Variable	
clinical	laboratory values
age	hemoglobin
sex	white blood cell count
AML type, de novo	platelet count
AML type, secondary	bone marrow blast count
AML type, therapy-related	peripheral blood blast count
extramedullary disease	LDH level
molecular genetics	
ASXL1	JAK2

BCOR	KIT
BCORL1	KRAS
CBL	NOTCH1
CEBPA	NPM1
CEBPA, monoallelic (TAD)	NRAS
CEBPA, monoallelic (bZIP)	PHF6
CEBPA, double-mutated	PTPN11
CSF3R	RAD21
CUX1	RUNX1
DNMT3A	SF3B1
ETV6	SMC1A
EZH2	SMC3
FLT3-ITD	SRSF2
FLT3-ITD ratio	STAG2
FLT3-TKD	TET2
GATA2	TP53
IDH1	U2AF1
IDH2	WT1
IKZF1	ZRSR2
	
cytogenetics	
	
Karyotype, complex	inv(3) or t(3;3)
Karyotype, neither normal nor complex	-5 or del(5q)
Karyotype, normal	-7 oder del(7q)
t(8;21)	-17 or del(17p)
inv(16) or t(16;16)	

Table 1 Variables used for cluster formation. In order to avoid the 'curse of dimensionality', i. e. the destabilizing nature of high-dimensional data sets with only limited data points in them, a feature had to be present in at least 1% of patients to be taken into account by the unsupervised model. Pre-processing of data using this cut-off allowed for stable computations without noise from underrepresented features that would otherwise

distort the distribution of data across n dimensions for n features. Further, apart from the 'curse of dimensionality' high values for n make the computation rather inefficient while likely not adding value if only sparse features (<1% present in the patient cohort) are included. The list depicted above constitutes the variables used for cluster generation. Each variable is weighted equally, i. e. no manual interference was undertaken to put additional weight to any of these variables which may potentially have introduced bias and would have been diametrically different from a data-driven rather than hypothesis-driven approach. Lastly, outcome variables were explicitly excluded from cluster generation and only information upon initial diagnosis was used stemming from clinical, laboratory as well as molecular and cytogenetic investigations.

With regard to 'minimal' features or rather the questions of whether all features are needed which implies whether there are features of higher or lower relevance, we want to point out that the mathematical set up of this unsupervised learning model as described above does not allow for any weight adjustments for individual features. First of all, this is not possible from a mathematical/technological standpoint: Each feature is essentially an axis in an n -dimensional coordinate system. A method to individually adjust weights for each axis is, at least to our best knowledge, non-existent. Further, at least from our perspective, this would also not be desirable. The explicit technological goal of our study is to have a model with minimal human interference and therefore, minimal hypothesis-driven bias. If we as MDs were to, for example, state that *NPM1* mutations are of high relevance and should therefore receive a different weight than any other given mutations, this would dramatically distort the output shape of the data while at the same time violating the data-driven set-up by introducing a manually drafted element of anticipation that 'X is more important than Y' without necessarily being true. As stated above for sAML, different variables and different data sets can potentially provide a different output shape in any unsupervised learning task. In our model, all 61 variables are necessary and all are weighted equally.

In order to clarify this peculiarity of unsupervised learning and dimensionality, we revised the Methods with regard to data pre-processing and transformation and added the following paragraph:

"Features used for cluster generation were available upon initial diagnosis and were either clinical variables (such as age, sex etc.), laboratory variables (such as Hb levels, platelet and white blood cell count etc.) or cytogenetic or molecular genetic alterations comprising a total of 212 parameters. Table S5 (revised version) shows a full list of variables and their frequencies in the patient cohort. Further, it has to be noted that in an unsupervised setting, a number of n features is transformed into n axis of a coordinate systems which represent the model space. To reduce model dimensionality, variables that were present in less than 1% of the patient cohort were excluded from analysis. This is intended to tackle the so-called 'curse-of-dimensionality', where computation is destabilized by adding dimensionality in a data set with a smaller number of samples compared to the available number of features. After excluding sparse features in order to

make computations more stable and efficient, 61 features were left. Importantly, no outcome features such as achievement of CR or survival times (EFS, RFS, and OS) were used for cluster generation. These outcome features were explicitly excluded from cluster generation as they may substantially bias cluster assignments. As individual features represent axis in a coordinate system, weights between features are uniformly equal and cannot be modified externally, i. e. each feature for cluster generation is as important as any other feature. Modifying features, either by inclusion of novel features or exclusion of present features, will therefore obviously modify the shape of the intermediary model and hence the results.”

lines 402 - 419

7. It seems that the resolution of cluster by survival according to KM was a criterion (sanity check) at some point. This however may impair the functional relationships with clusters: was the best resolution of survival a criterion? As such one would think that e.g. short survival may be function of various not disease related factors and treatment. E.g., patient who died due to sepsis or car accident early may have totally different genetics..... or patients with distinct genetic profiles may have similar survival. This needs to be clarified and discussed. What was the overall aim of this analysis.

11. The authors combine multiple transformation/dimension reduction methods with clustering. However, not all of the transformations described are suitable for different types of inputs. For instance, principal component based methods assume the signal to follow a Gaussian distribution for which categorical variables/one-hot encoded variables are not appropriate. Similarly, not all of the transformations are applicable for the utilized clustering methods. Non-linear transformations are better suited for density based clustering (DBSCAN) vice versa hence an explicit weighting might be required during aggregation which is not described as well.

We agree that the sanity check could be described in more detail. First of all, as we stated in response to the reviewer’s comment 4, survival variables were by no means part of the cluster generation. The purpose of cluster generation, of course, is to find clusters that biologically differ and provide a meaningful impact on clinical courses and outcomes. The sanity check did not consider survival. Neither did the clustering algorithms overall. Its sole purpose is to check for quantitatively measurable parameters of cluster quality. That means, for example, that in theory it would be possible for clustering to generate 1383 clusters, i. e. one cluster for each patient. It would also be possible to generate one giant cluster for all patients. Therefore, some rules have to be introduced in order to make clusters somewhat interpretable. The rules we used were simply quantitative measurements: number of individual clusters, relative cluster sizes, and cluster differences. We did not expand on this and why we chose specifically these in the previous iteration of the manuscript.

We agree with the reviewer: Not all combinations are well suited for each other. This is another reason why the sanity check comes in handy. If the distribution of the data is not well suited for any given transformation step among the eleven available algorithms, the shape of the data will be extreme in some form. Also, if the clustering for any given previous transformation is not well suited, the individual output will also be extreme in some form. As described in response to comment 7 regarding the sanity check, these extreme results will be filtered out. Rather than analyzing 11 x 11 combinations by hand to see whether or not anything will not match, we just screen them all and eliminate extreme results as described in comment 7. Again, thereby we eliminate any assumptions we have to make manually and rather let the data speak for themselves.

We added the following paragraph to further explain the function and purpose of the sanity check:

“To ensure optimal quality of the clusters, a sanity-check was performed after each clustering. This is intended to make clusters both interpretable and clinically meaningful. From a purely mathematical standpoint, it would be conceivable that cluster generation would end in 1383 clusters, i. e. one cluster for each patient, or one cluster for all patients as well as any number of clusters in between. As this is not practicable to be used in the clinical routine, a set of rules has to be introduced to ensure that generated clusters fulfill a set of criteria that enables their transferability from computer to bedside. To highlight biological differences between AML patients, clusters have to differ maximally in their data distribution. In an n-dimensional space, this means that each patient is represented by a point in a coordinate system with n axis. Cluster generation should then be able to delineate clusters by maximizing the distance between patients in the n-dimensional space. This means that only patients that are close to each other with regard to their features should be considered belonging to the same cluster while patients that are different, i. e. are farther apart in this n-dimensional space, should belong to a different cluster. Further, clusters should be of adequate size to be meaningful in clinical routine. A cluster that consists of only e. g. five patients (of 1383 patients in total) would not be meaningful. At the same time, such small but distinctive clusters would dramatically increase the overall number of clusters and thereby further hinder clinical applicability as it seems rather improbable that a clinician would utilize (or even memorize) a risk assessment tool consisting of a two- or even three-digit number of individual risk groups. Therefore, cluster sizes and numbers of clusters (minimum of 10% of overall patients, i. e. limiting cluster numbers to a maximum of 10) were also considered as rules in the sanity check. Since we aimed to include a variety of transformation and clustering algorithms rather than subjectively selecting any specific combinations, it has to be pointed out that depending on the distribution of the data not all transformation algorithms are suitable to work with all kinds of data and not all combinations of transformation and clustering algorithms work well with each other. As our explicit goal from a technological perspective was to manually interfere as little as possible and rather let the model decide for itself and eliminate unfitting outliers, we nevertheless allowed all 11 x 11 combinations of transformation and clustering algorithms. Conceivable, a dysfunctional

match-up between any of these algorithms would produce a result that would be extreme in some sense, e. g. would produce to large or small, to few or to many clusters, or patients within the clusters would be too different and patients between the clusters would be too similar. Again, this is where the sanity check comes in that eliminates such extreme and mathematically unsound combinations before a final analysis is undertaken. Thereby, we do not need to sort through combinatorial outputs manually and potentially introduce subjectivity, but rather let the model decide under the pretense of the above-specified rules for clustering which clusters and thereby which algorithmic combinations satisfy the pre-specified quality criteria. It needs to be stressed, that the sanity check did in no way interfere with feature selection nor did it include any outcome variables to maximize cross-cluster heterogeneity. Outcome variables were strictly withheld from cluster generation and only statistically analyzed after final clusters were generated. If an individual run fails the sanity-check, for example if it includes clusters harboring only 1 patient each, the result is discarded and not evaluated further."

lines 470-507

8. The quality of prognostic prediction DFS, OS etc... could be assessed e.g. by C scores estimated vs actual survival since the analysis is retrospective

It has to be stressed that there is no prediction here as in supervised learning! C-Scores could be used in a regression or prediction problem in supervised learning where a synthetic value is predicted based on baseline patient features. However, in our set up, all survival times are the real survival times the patients actually had! There is no prediction! Unsupervised learning essentially sorts the patients into groups, nothing more. Hence, it was not necessary to estimate survival, because all survival times for the patient were already available. The differences in survival times between the clusters are grounded in the actual differences in survival between patients assigned to each cluster.

9. The treatment modalities in each cluster should be shown. How many patient received what?

All therapies were intensive regimens. We added a table to the supplements to clarify this for the respective clinical trials:

trial name	clinicaltrials.gov identifier	trial duration	protocol summary
AML96	NCT00180115	1996-2008	risk-adapted postremission treatment regarding allogeneic stem cell transplantation for high-risk AML and related allogeneic

AML2003	NCT00180102	2003-2009	and autologous stem cell transplantation for standard-risk AML, and randomization between intermediate-dose and high-dose cytarabine within the first post-remission course early allogeneic stem cell transplantation in post-induction aplasia for high-risk AML, factorial design with four therapy arms with two factors of two stages (intensified vs. standard therapy and cytarabine vs. cytarabine + mitoxantrone + amsacrin)
AML60+	NCT00180167	2005-2010	Patients ≥ 60 years, mitoxantron on day 1,2,3 + cytarabine on days 1,3,5,7 vs. DA 7+3
SORAML	NCT00893373	2011-2014	Standard therapy + sorafenib vs. standard therapy + placebo
SAL bioregistry	NCT03188874	2010-present	Prospective registry of AML patients
AMLCG-1999	NCT00266136	1999-2007	double induction with HAM-HAM, multiple course G-CSF or myeloablative consolidation with Bu/Cy and autologous blood stem cell

AMLCG-2008	NCT01382147	2008-2012	transplantation instead of maintenance vs. standard therapy S-HAM escalated for younger patients and S- HAM basis for elderly patients vs. TAD-HAM (younger) or HAM- HAM (elderly)
------------	-------------	-----------	---

10. The authors describe a combinatorial workflow of 605 unsupervised clustering methods. However, important details regarding the selection of the methods, optimization of parameters and aggregation of clustering results are missing. For instance, the authors utilize both the PCA and SVD which are indistinguishable in practice and result in the same projections. Similarly, AE model is not described, nor are the parameters involved (number of layers/units of layers etc.).

As the reviewer correctly points out, some algorithms are somewhat similar while other substantially differ in their functionality. This leads to the essential question of which algorithm is to be chosen. In the literature, we usually see only one algorithm being reported for any given task. Often, it is unclear whether actually only one algorithm was tested or whether multiple algorithms were tested and only (for whatever reason) was reported. Our approach explicitly aims to combat that as there is no universal ‘best’ algorithm and the performance of a model is heavily influenced by input data and task. We aimed to include a large variety of both transformation and clustering algorithms in order to eliminate human bias in algorithm selection. Subsequently, when using such a large combinatorial approach, the question remains: Which combination performed best?

Again, in supervised learning this question would be trivial to answer. Each algorithm would get a shot at predicting a set of solid ground truth labels in a test set and the best performing algorithm would simply be the one with the highest overall hit-rate. Contrastingly, in unsupervised learning this is much more difficult as no labels are given to evaluate accuracy on. We heavily discussed this during the design of this study. One could arguably perform statistical analysis for all 605 possible combinations and compare outputs side by side. However, one would still need to pick one of these outputs over all others. This would be largely biased by a human observer. Again, this is what we explicitly set out to minimize in our study design. Rather than picking one ‘winner’ combination – although the criteria of what constitutes a ‘win’ with an uncertain ground truth is rather fuzzy – we simply averaged all results using meta-clustering. This way, we can eliminate any manually introduced bias in selecting a ‘winning’ model and rather just incorporate

all outputs for subsequent analysis. The fact, that on a more granular level not only outcomes of the individual clusters but also a large proportion of individual features (see Table 2 and Figure 2) differ significantly, serves as a proof of concept.

We agree with the reviewer that a more detailed appreciation of this train of thought is warranted. We added the following paragraph to the Methods:

“Inherently to unsupervised learning, different methods of clustering, i. e. different algorithms, will result in different outputs. Hence, no single ‘best’ algorithm exists for any given task, but rather the algorithm has to be evaluated within the context of a specific data set. What further complicates this issue, is that there are no labels in unsupervised learning. A supervised learning task is trivial to evaluate with regard to differential model performance: A given set of robust ground truth labels are provided and model performance is simply evaluated on how well each single model predicts the previously unseen labels. Then, the best model is the one with the highest hit-rate between predicted and ground truth labels. Since there are no labels (and often no real ground truth) in unsupervised learning, a selection of the ‘best’ algorithm is rather subjective. In our set-up of 605 different possible combinations, one could conduct statistical analysis for each combination, but would still be left with the issue of what output would be considered the ‘winner’. As this fundamentally ends in a subjective and therefore potentially biased manual decision, we used meta-clustering not to select the ‘best’ clustering output, but rather to average all valid (after the sanity check) outputs. Meta-clustering was performed using principal component analysis for transformation and mean shift for clustering. Meta-clustering essentially does not cluster the raw data, but clusters the output of the previous algorithms (Figure 5). Thereby, the individual outputs of all 605 combinations are used to generate clusters based on similar cluster assignments, i.e. patients that are farther apart in the n-dimensional model space for the majority of clustering algorithms will also be farther apart in meta-clustering (on average) and will therefore be put in two different final clusters. In that way, the chance for false positives or negatives that may be given when using only one combination of transformation and clustering algorithms is reduced and the need for human judgement of what makes one combination better or worse than another (and thereby potentially introduce bias) is limited. ”

lines 508 - 540

With regard to specifics of model parameters, we cannot fully provide in depth descriptions of each algorithm as we are limited by editorial constraints regarding space. We mainly used the standard functions of the scikit-learn package for Python which are specified in the respective scikit-learn documentations. Specifics with regard to code and its applicability as well as native settings we used for the purpose of this study are fully provided in the GitHub repository we linked in the Code Availability section with full access to all code developed for our experiments, e. g. settings for autoencoders.

12. The authors describe no validation approaches utilized for the identified clusters. One straightforward approach would be to do a k-fold cross-validation at least with a subset of the combinatorial workflow.

For external validation, please see the comments above. As stated in lines 467-469, for original clustering each run was repeated $k=10$ -fold.

13. Finally, I find the conclusions modest, ie: essentially AML is more complex than ELN can appreciate". A list of concrete new results findings should be provided.

We respectfully disagree with the reviewer with regard to the modesty of our study's results. The ELN recommendations are probably the most widely used system of risk stratification in the EU (and beyond). We explained in detail above that the decision-tree-like structure of risk assessment via ELN likely fails to acknowledge disease complexity. This at least suggests that a substantial number of patients could receive a more granular individual risk assessment rather than just hitting an item on the ELN checklist and being categorized into a heterogenic group of patients with somewhat subjectively considered similar risk. This is represented in the fact that all three ELN groups are prevalent in varying proportions in all four of our clusters. Potentially, this suggests that ELN-based treatment decisions that we regard as a given in clinical routine could be challenged by data driven models. However, an unsupervised retrospective study design is not suitable to address this. Rather a data-driven reinforcement learning approach, ideally on a synthetic data set in an online (i. e. synthetically prospective) setting would be a suitable alternative. This, however, is obviously not within the scope of our study.

We added to the discussion: "The lessons from this study are threefold in terms of i) acknowledging disease biology and interconnectedness of genetic alterations in a combinatorial model, ii) providing a data-driven tool for cohort-based risk assessment with clinically meaningful differences in patient outcome that potentially warrant different disease monitoring strategies, and thereby iii) challenging conventional approaches for risk stratification that are expert-opinion-based rather than data driven."

lines 230-235 and the following paragraphs. Each of the points summarized above is further expanded upon within the revised discussion and partially addressed in previous responses.

While our clustering results are not prospectively validated and subjected to limitations such as the ones regarding a predominantly central European population etc. as stated above, we believe our data-driven technological approach, if used in an iterative manner on a growing cohort of patients, can provide a more nuanced tool to risk stratification. Of course, this is not a final tool (as no unsupervised-learning-based tool can arguably ever be termed final as addition of novel data always warrants an update). Our results, however, point into the direction, that re-iterating an expert-based checklist every five

years is likely not well suited for the clinical routine given the rapid progress of basic science in generating a better understanding of disease biology in conjunction with the ongoing digitization of medical data in clinical routine. This would constitute a paradigm shift to clinical management with AML being a model disease where previous expert panels are abandoned in favor of large-scale multinational data driven approaches. While we are far from the latter, we still consider our study (among a variety of others) as a proof-of-concept.

Minor Comments

1. Regarding the Figure 3, KM curves, confidence bands for individual curves and the number of samples at risk should be given.

We agree and have updated Figure 3 accordingly. Please see the corresponding figure above in our response regarding external validation.

2. It would be informative if overlap of identified clusters (e.g. adjusted rand index) across different workflows were shown in order to quantify the utility of using multiple workflows better.

Thank you for this comment! Yes, we agree that some sort of measurement would be informative to evaluate the performance of individual algorithm combinations. Evidently, it is hard to measure what 'good' clustering is. Arguably, clustering has to serve a translational purpose, i. e. separate patients according to high- or low-risk features, differing disease biology etc. There are a number of scores and indices that can be used to evaluate cluster performance, but these are of limited information with respect to meaningful knowledge gains. For example, if we took the silhouette coefficient (ranging from -1 to +1) and separated clusters only by sex as a feature, we would likely get silhouette coefficient very close to 1 (since at least in our cohort this feature is reported as binary). However, the literature suggests that there is no difference in male vs female patients with regard to AML outcome. Now, we would have a very high silhouette coefficient that looks good on paper, but we would not have any gained information, differences in outcome etc. So, these indices have always to be considered in context with knowledge abstraction from clustering data. A good numerical score does not necessarily mean that we will get good information retrieval from the data and vice versa.

Nevertheless, we agree that some sort of numerical scoring system is relevant to compare individual algorithm combinations. We therefore calculated three different indices for each combination: silhouette coefficient, Calinski-Harabasz-index, Davies-Bouldin-score. We added the following paragraphs to the Methods and Results:

Methods:

“Still, to make individual algorithmic combinations numerically comparable, we used the silhouette coefficient, the Calinski-Harabasz-Index, and the Davies-Bouldin-Score. Silhouette analysis measures how close a point in a cluster is to points in neighboring clusters. On a scale from -1 to +1, samples that are far away from neighboring clusters will receive a value close to +1 while a value close to 0 means that a sample is close to the decision boundary between two clusters whereas a value close to -1 indicates an error in cluster assignment. The Calinski-Harabasz-Index (also known as the Variance Ratio Criterion) is the ratio between the sum of inter-cluster dispersion and the sum of intra-cluster dispersion. It ranges from 0 to (theoretically) infinity with higher values indicating higher clustering quality. Lastly, the Davies-Bouldin-Score is an average similarity index that compares each cluster to its most similar cluster. A ratio is formed of intra-cluster distances to inter-cluster distances. The score ranges from 0 to (theoretically) infinity where values close to 0 indicate a higher distance between clusters and thus, better overall cluster quality.”

lines 520-531

Results:

“These features were solely comprised of clinical variables such as age, AML status (de novo, sAML, tAML) etc., laboratory variables such as bone marrow blast count or white blood cell count etc. upon initial diagnosis, and molecular and cytogenetic variables. For a comprehensive list of all 61 features included within the model, please see Table S1. Individual clustering performance of different combinations of transformation and clustering algorithms as well as target dimensions were evaluated using the silhouette coefficient, the Calinski-Harabasz-Index, and the Davies-Bouldin-Score. Silhouette coefficients, Calinski-Harabasz-Indices and Davies-Bouldin-Scores ranged from -0.12 to 0.72 (closer to 1 is better), 4.56 to 3533.94 (higher is better), and 0.46 to 18.46 (closer to 0 is better), respectively. 517 of 605 possible algorithmic combinations passed the sanity check. Their individual performance metrics can be viewed in Table S2. Since a good numerical value in the above-mentioned indices does not necessarily guarantee meaningful knowledge retrieval from clustering and picking a ‘winner’ combination still remains somewhat subjective, meta-clustering was used to average the results of combinations that passed the sanity check.”

lines 96 – 108

We also added the table of algorithm combinations and their individual performance metrics to the supplements (please see Table S2)

3. Discussion is very dense

We agree that it is a lot to unpack. The topic itself and especially the methodology is complex, however, we want to provide the reader with a detailed discussion both

through a medical and technological lens. We have also edited the discussion thoroughly according to both reviewers' suggestions and hope the updated version sheds light on some thought processes thereby reducing the density of different arguments in the previous version.

Reviewer #2 (Remarks to the Author):

I am happy to sign this report to the authors. I am David Westhead from the University of Leeds.

I think this is an interesting piece of work, but I would raise some issues that I think need attention.

1. First and foremost, I think that to have substantial influence on the field this work needs to be pushed a bit further. I would like to see some effort at validation, ideally in an independent cohort of patients, rather than just a clustering of a single cohort. In the absence of a suitable independent cohort it would in this context have been useful to hold out a validation/test from the data set used. There is always a risk that biases in any one cohort lead to methods that don't generalise well in different cohorts, and also that data analysis decisions introduce cohort dependent bias in method and parameter selection. Since this data is already a union of cohorts of patients from different sources you might consider whether using just one of those sources as test/validation would be a good option?

You comment in the discussion that prospective evaluation would be needed, and for this you will need a method to assign a new patient to one of your clusters, i.e. a classifier. Developing this for the final clustering solutions in the training data set would then give a way of assigning cluster membership in the test/validation data. And with that you could check the expected associations of cluster membership with response and survival in the validation/test data. This would strengthen the paper substantially.

First of all: Thank you for the kind words and the excellent suggestions! We completely agree with the reviewer that external validation may help in solidifying results. As explained in detail above and in response to reviewer #1's comments (see response to comment no. 3 of reviewer #1 in particular), external validation is not a trivial task in unsupervised learning.

Just briefly, without repeating too much with regard to the external cohort that has already been answered in the responses above, we obtained a very similar cohort of 664 intensively treated AML patients from another multi-center collaborative network with very high overlap in available features. Please see the updated Methods for reference:

“For external validation of an unsupervised learning task, the availability of overlapping features between an original cohort and an external cohort is essential since the distribution of the data shape is tightly connected to the features for any given data set. Hence, a data set that diverges substantially in its available features will inevitably produce different results in an unsupervised learning task. We obtained another large cohort of 664 intensively treated AML patients for whom the same eligibility criteria applied and for whom the same features were available (with the exception only of IKZF1 and FLT3-TKD mutation status). This cohort was provided by the AML Cooperative Group, a multicenter collaborative group of university centers from southern Germany and Austria, and was comprised of patients treated within previously reported clinical trials (AMLCG-1999 and AMLCG2008). Specific treatment regimens for all different trial protocols are summarized in Table S3.”

lines 378-388

We really want to thank you for the excellent insight of turning the issue into a classification problem! We did exactly that. As you can see in the updated Figure 5, we used cluster assignments as labels and trained a supervised classifier to predict cluster assignments on the external cohort.

Figure 5. Step-wise workflow of unsupervised learning. After pre-processing of multimodal patient data (1), unsupervised learning (2) was performed with different combinations between target dimensionality (2 – 6, $n=5$), data transformation ($n=11$) and

unsupervised clustering algorithms (n=11). The individual outputs of each algorithm combination were gathered and used as input for meta-clustering (3) to find patient clusters that in themselves are maximally homogenous while at the same time differ maximally from other patient clusters. Thus, final clusters are identified and individual features of patients within these clusters become available for further analysis. In the next step, the previous cluster assignments can be treated as labels for supervised learning (4). After training and testing on the original cohort, the highest performing classifier is being selected for assigning cluster labels to an external validation cohort.

We updated the Methods and Results accordingly. Since reviewer #1 also requested an external validation cohort, we would like to point to the response above. Please also see the revised Methods, lines 402 – 437, 470-531, 543-571 and results, lines 201 – 223.

2. Your treatment of missing data is a concern. Exclusion of variables present of <1% of the patients is a relatively weak condition and leaves other variables in with significant missingness. The approach of just replacing with the median needs to be considered. Missingness may not be completely at random, and there is a danger that with naive treatment you can inadvertently code other information about the patients. For instance, if doctors order tests on the basis of some prognostic information then missingness of these tests can contain prognostic information. Machine learning methods can discover this and learn patterns that are not useful, and it could contribute to the observed survival/response differences between these clusters. I would recommend a more careful treatment of missing data, considering these effects and evaluating as well more sophisticated methods of multiple imputation, as well as complete case analysis if feasible.

We thank you for this comment and raising this important issue! You are absolutely right: Missingness of a test may itself be seen as a kind of information and verification whether or not data is missing completely at random or missing not at random is important not only for choice of imputation method, but rather for internal validity itself.

In the previous version of the manuscript, we were rather cryptic with regard to how missing data was handled and the two lines we provided fell short of explaining the system well enough. First of all, since data came from previous clinical trials, and since we analyzed molecular alterations for the entire cohort, data quality itself should be sufficiently high. For full transparency, we provide a list of missing values for the variables that were included in cluster generation, Table S6:

Variable	n missing (%)
clinical	
age	0

sex	0
AML type	17 (1.2)
extramedullary disease	137 (9.9)
laboratory values	
hemoglobin	6 (0.4)
white blood cell count	2 (0.2)
platelet count	4 (0.3)
bone marrow blast count	103 (7.4)
peripheral blood blast count	79 (5.7)
LDH level	65 (4.7)
molecular genetics	
ASXL1	0
BCOR	0
BCORL1	0
CBL	0
CEBPA	0
CEBPA, monoallelic (TAD)	31 (2.2)
CEBPA, monoallelic (bZIP)	31 (2.2)
CEBPA, double-mutated	31 (2.2)
CSF3R	0
CUX1	0
DNMT3A	0
ETV6	0
EZH2	0
FLT3-ITD	0
FLT3-TKD	0
GATA2	0
IDH1	0
IDH2	0
IKZF1	0
JAK2	0
KIT	0
KRAS	0
NOTCH1	0
NPM1	0
NRAS	0
PHF6	0
PTPN11	0
RAD21	0
RUNX1	0
SF3B1	0
SMC1A	0
SMC3	0
SRSF2	0
STAG2	0
TET2	0
TP53	0
U2AF1	0
WT1	0
ZRSR2	0
cytogenetics	
Karyotype, complex	81 (5.9)

Karyotype, neither normal nor complex	81 (5.9)
Karyotype, normal	81 (5.9)
t(8;21)	81 (5.9)
inv(16) or t(16;16)	81 (5.9)
inv(3) or t(3;3)	81 (5.9)
-5 or del(5q)	81 (5.9)
-7 oder del(7q)	81 (5.9)
-17 or del(17p)	81 (5.9)

Table S6 Missing data for variables included in cluster generation.

We traced these back as best as we could and the most likely reason for missingness is simply a lack of documentation, e. g. for the 81 missing karyotypes. There is no patient for whom information is uniformly missing, which could be the case if there was some kind of systemic error. Since this is not the case, we assume missingness completely at random since every patient has the same chance to have their values not documented.

We discussed this in depth and we also agree with you that in most cases more sophisticated methods of imputation are usually the go-to method such as multiple imputation as you suggest. However, the problem here lies within the reproducibility of results: Due to randomness in imputation likelihood, multiple imputation will provide different imputed values every time the imputation is run. This is a bigger problem than using simpler methods of imputation under the MCAR assumption in our set up. The reason is that introducing an element of randomness via multiple imputation will completely abrogate the workflow in terms of subsequent analysis of clustering results. Unsupervised analysis in general is most commonly a 'black box'. While for supervised models, different methods of explainability exist, unsupervised models commonly provide a result that has to be sort-of accepted the way it is without finding a reason why the model 'decided' the way it did. Having a simple method of imputation makes this workflow at least reproducible. Having a non-reproducible method of imputation that produces a different result for the entire workflow every time. This means that first of all we could not run the analysis multiple times and get the same results which would somehow challenge the internal validity of the clusters itself. Plus, if (God forbid), any data would be lost due to technical issues, memory crashes etc., the entire analysis would be invalid, since the workflow would have to start anew from scratch since the original cluster creation would likely not be reproducible.

Long story short: We are aware that simple imputation is not optimal, but it is the only way (in our perspective) to create a robust workflow.

Plus, as is evident from the table above there are not that many missing values after all. Missing ordinal values were essentially dropped from the patient's feature vector (see below) and hence, the consideration above only applies to missing lab values. Again, no outcome variables (CR rate, EFS, RFS, OS) were incorporated into cluster generation! Kaplan-Meier analysis was conducted only for patients with available values for each dependent variable. This was also a bit misleading in the previous version of the

document, but becomes clear by providing number-at-risk-tables as suggest by reviewer #1.

Exclusion of variables that were present in <1% was necessary due to the discrepancy between overall available variables and overall available samples (=patients). In a much larger cohort, analysis with more variables could be feasible, however, in this cohort, although large from a clinical point-of-view, would likely destabilize the model. Please see our response to reviewer #1's comments 5 and 6 with regard to the so-called 'curse of dimensionality'.

We added the following paragraph to the methods in order to make our though process clearer:

"Many statistical and machine learning models do not cope well with large amounts of missing data which may ultimately result in unstable or biased models depending on the mechanisms of missingness. A full list of absolute and relative numbers of missing values for the features included in the model is provided in Table S5. Since unsupervised clustering itself poses a 'black blox'-like dilemma with regard to explainability, introducing a multiple imputation mechanism that generates different results each time an imputation is run would not be a suitable option, if one aims at reproducible results. No missing data was present for age, sex and molecular alterations (with the exception of subtyping for CEBPA mutations into specific domains in 31 cases), and only a fraction of karyotypes (5.9%) and laboratory values (range 0.2 – 7.4%) were missing. In order to generate reproducible outputs for imputation that would still be solid in k-fold cross-validation and potential re-runs of the model, continuous variables were imputed with the median of the respective variable. Missing categorical variables were tagged as unknown. Unknown variables were dropped at the pre-processing stage. For example, a patient with all data available except for information on extramedullary AML manifestations will still be included in clustering, however will be represented by a feature vector consisting of 60 values rather than 61 in a 61-dimensional intermediary model space. With regard to outcome variables, which were strictly withheld from cluster generation, only cases with available survival times were used in Kaplan-Meier analysis (see numbers-at-risk-tables, Fig. 4). There was no imputation of missing outcome variables."

lines 420 - 437

3. In considering differences in variables between your clusters I would refrain from attaching p values to variables that are used in clustering. These are not really meaningful: by clustering on a variable and then comparing patients in different clusters you are selecting groups of patients that are already expected to differ on the clustering variable. That means that the p values are not meaningful in the usual sense, and you could argue that they really only prove that your clustering is working.

The overall meaningfulness of p-values is probably a philosophical debate, but we agree with you. Still, it would be possible that cluster differences were only brought forth by a small number of variables while the majority of variables could be of low relevance to cluster generation. Since there is no real method of weighing the influence of each variable on cluster generation (see our response to reviewer #1's 2nd comment), this at least provides the reader with the information that not only a handful of variables were important for cluster generation, but the overall combination of information is important since a lot of variables turned out to significantly differ between clusters. Of course, you are right that this is somewhat expected by optimizing for this during cluster generation. We only argue that the extent to which this criterion is fulfilled cannot be anticipated a priori. As you mention, this only serves as a proof-of-concept that cluster generation is working adequately. This in and of itself is still worthy to be noted by the reader, we believe.

4. The meta clustering algorithm needs detailed explanation for reproducibility. It is important to understand how the eventual solution of 4 clusters is arrived at and in what sense it is optimal.

Again, we agree and apologize for not providing enough detail in the first place. Reviewer #1 also commented on this in several comments. Please see our responses to reviewer #1's comments 5, 6, 7, 10, 11. We completely revised and expanded the sections on 'Data pre-processing and dimensionality reduction' as well as 'Unsupervised learning and meta-clustering', lines 401 – 542. We hope the updated version is to your satisfaction.

5. Further to point 4, the ideal solution to reproducibility is to provide a script (or perhaps a Jupyter notebook or similar) that does all the analysis from raw data input to output figures and tables for the paper. I know this is hard work, but I think it is something to which we should aspire, and it saves work for everyone who wants to use or extend the work in the future.

We agree that reproducibility is an important issue. We therefore provide the code that was generated for the purpose of this project so that anyone can download it via GitHub and tailor it to his or her needs. While our unsupervised learning and meta-clustering pipeline is relatively generic when it comes to re-purposing it for different tasks than the one at hand, our pre-processing pipeline is not. The workflow is very much tailored to our original data set and the external validation cohort, which largely overlap. We want to point out that even large commercial enterprises still struggle to provide an easy-to-use software solution that can provide no-code solutions for machine learning for a variety of different data set/type inputs. As one example, SPSS modeler comes to mind that is still very rigid in data entry and format requirements although being released by one of the biggest IT companies in the world (IBM). We gladly provide our code for anyone to use, but building a multi-purpose data inflow pipeline that can be used for any data set, especially raw data wherever it may come from, is unfortunately out of the scope of this

work in terms of (wo-)manpower and financial resources. Furthermore, the output of the code is largely in tabular form with the exception of Figure 1. The other figures were generated using statistical software, Microsoft Excel and Powerpoint.

The code is provided under the following link:

<https://github.com/sit-institute/sal-metaclustering>

6. Finally, the presentation of methods after results does mean that in places you need to give just a bit more information in the results section. Some statements are hard to interpret without reading ahead to methods, e.g. 'aggregated the results of 605 different combinations of target dimensionalities' needs just a bit more local explanation to be meaningful to read. I can probably find other examples.

Again, we completely agree with you. However, due to editorial constraints regarding the structure and word count this is rather challenging. We substantially revised the Methods section in order to make a lot of concepts clearer to the reader. The required structure, however, is first Results, then Methods. Repeating methodology in the results is difficult to do in only a few words since we apparently fell short to comprehensively describe a couple of concepts in the previous version by using rather short statements for which we sincerely apologize. We are somewhat worried this would also be the case if we were to just add a few words here and there with regard to methodology to the Results.

Obviously, we want the reader to understand the workflow in detail rather than just get an idea of it superficially. Therefore, we would actually prefer the reader to have a glance at the Methods first and hence, it is probably more confusing to add short repetitive statements to the Results.

REVIEWERS' COMMENTS:

Reviewer #2 (Remarks to the Author):

I think this is a commendable response to address the comments of both reviewers. I appreciate that the external validation was a lot of work but it does substantially enhance the contribution made by the paper.

Reviewer #3 (Remarks to the Author):

Thank you for the opportunity to review this paper. This is an important piece of work that uses unsupervised learning to identify patient clusters of prognostic importance. The work is interesting and clinically relevant. Technically, the work is well performed and the conclusions are appropriate.

I have specifically evaluated the response to Reviewer 1 comments, as requested.

I believe that the authors have adequately and appropriately responded to all of the comments raised within. Specifically, the addition of a validation cohort appears to have been an important issue for both reviewers, and this has now been addressed.

The additional data points and clarifications that were requested have now been provided.

The additional impact of these findings on prognostic indicators is well appreciated .

I have no further specific comments or concerns.